# A statistical framework for high-content phenotypic profiling using cellular feature distributions

Yanthe E. Pearson [1], Stephan Kremb[1], Glenn L. Butterfoss[1], Xin Xie [1], Hala Fahs[1] & Kristin C. Gunsalus [1,2]✉

High-content screening (HCS) uses microscopy images to generate phenotypic profiles of cell morphological data in high-dimensional feature space. While HCS provides detailed cytological information at single-cell resolution, these complex datasets are usually aggregated into summary statistics that do not leverage patterns of biological variability within cell populations. Here we present a broad-spectrum HCS analysis system that measures image-based cell features from 10 cellular compartments across multiple assay panels. We introduce quality control measures and statistical strategies to streamline and harmonize the data analysis workflow, including positional and plate effect detection, biological replicates analysis and feature reduction. We also demonstrate that the Wasserstein distance metric is superior over other measures to detect differences between cell feature distributions. With this workflow, we define per-dose phenotypic fingerprints for 65 mechanistically diverse compounds, provide phenotypic path visualizations for each compound and classify compounds into different activity groups.

[1] Center for Genomics and Systems Biology, New York University Abu Dhabi, P. O. Box 129188, Abu Dhabi, UAE. [2] Department of Biology and Center for Genomics and Systems Biology, New York University, New York, NY 10003, USA. ✉email: kcg1@nyu.edu

High-content screening (HCS) is an easily automated and cost-effective tool to generate rich image-based datasets that capture a wide variety of cellular phenotypes. High-dimensional numeric feature sets are then extracted from images to generate phenotypic profiles that characterize cytological responses to chemical or genetic perturbations. Image-based cytological profiling has gained significant momentum over the last two decades[1–3] for gauging the phenotypic impact of different treatments, inferring mechanism of action[4–10], identifying signatures of disease or toxicity, and characterizing cellular heterogeneity[11].

A central goal in HCS is to identify characteristic phenotypic responses that can be used to classify compounds with different cellular mechanisms of action (MOA). Best practices in experimental design such as placement of control wells, mitigating spatial biases across the plate, and the use of statistical metrics for phenotypic scoring have been discussed and reviewed[12]. However, no community-wide consensus has yet been established and the field is diverse in experimental design and choice of cell lines, biomarker probes, and compound doses applied[13–18]. In addition, published studies tend to use a limited set of probes based on fluorescent dyes or antibodies, which are usually combined into a single assay panel[17,19–21]. For example, the Cell Painting protocol[22] uses a single panel of six markers imaged in five channels. This simplifies the staining procedure, but also constrains the number and diversity of cellular features that can be measured. Using multiple marker panels[19,23] allows for surveying a broader spectrum of features and can reduce the risk of bleed-through between fluorescent channels depending on the assay design. While this adds experimental complexity and potential cost, an expandable set of cellular labels offers distinct advantages, particularly when there is no a priori target phenotype of interest and could become routine with advances in high-throughput imaging technology and analysis software[24].

The sheer quantity of high-dimensional single-cell data generated from HCS presents challenges to efficient analysis and data integration[25,26], and thus much of the data produced remains considerably underutilized. To date, ensemble measurements, such as mean, median, percent of control and standardized Z-scores tend to be the methods of choice for phenotypic profiling. Whether such aggregate estimators are sufficient, or too simplistic for characterizing phenotypic responses to perturbations, is not yet established and may depend on the biological system in question[27]. While the Z-score is commonly used to quantify phenotypic differences between treatment and control conditions[23,28], it oversimplifies interpretation and will fail to capture changes in the modality of population-level feature distributions or subpopulations with different responses[11]. As single-cell features (e.g., intensity, shape, texture) may exhibit diverse distributions, the exploration of alternative statistical metrics that are sensitive to arbitrary shape and size could be advantageous in detecting both subtle changes and skewed distributions.

Here, we describe a broad-spectrum HCS assay designed to maximize the range of detectable cellular phenotypes and used it to survey the sensitivity landscape of cytological responses to a small set of compounds with different reported mechanisms of action (MOAs). Our data handling and statistical workflow addressed several challenges of these data, with a focus on the following themes: position and plate effect detection, cell-level data standardization, statistical metric performance comparisons, feature reduction and broad-spectrum compound profiling. We further describe ways to characterize compounds based on cell counts, cell cycle distribution, and phenotypic dose responses, with practical visualizations of dose-dependent phenotypic trajectories in a lower-dimensional latent space. Our analytical framework enables the integration of feature measurements derived from multiple marker panels and provides a more comprehensive phenotypic overview of chemical perturbation that can be adapted to multiplexed HCS experiments with any set of reporters.

## Results

**Overview of experimental design, data acquisition, and analysis workflows.** In this study, we developed a broad-spectrum assay system (Fig. 1a–c) and companion analysis workflow (Fig. 1d–h) for high-content phenotypic profiling of mammalian cells. The HCS assay system was designed to maximize the number and diversity of cytological phenotypes that can be measured in response to chemical or genetic perturbations. It comprises commercially available fluorescent dyes and genetically encoded reporters that label ten different cellular compartments and molecular components, distributed across multiple fluorescent channels and assay panels: DNA, RNA, mitochondria, plasma membrane and Golgi (PMG), lysosomes, peroxisomes, lipid droplets, ER, actin, and tubulin (Fig. 1a, b).

Using automated high-throughput microscopy, images of each well were acquired and 16 cytological features were measured for individual cells for each marker in each of the four panels, for a total of 174 texture, shape, count, and intensity features (described in "Image acquisition and data extraction"). To harmonize and systematically integrate feature data stemming from multiple panels and different plates, the analysis pipeline (Fig. 1d–h) first detects and adjusts for positional effects, performs data standardization and statistical metric comparisons, and identifies the most informative features, which are then used to generate phenotypic profiles and visualize phenotypic trajectories in a low-dimensional space.

We tested the performance of this system by applying it to survey the bioactivity of 65 compounds with diverse MOAs and low structural similarity at multiple concentrations in human U2OS cells (Fig. 2a). Assays were performed in 384-well plates using a layout (Fig. 2b) with a total of 55 control wells distributed across all rows and columns (red) and a dilution series of each compound at seven concentrations (blue). Three technical replicates were performed for each of the 65 compounds, which were distributed across two plates (32 and 33 compounds per plate) per replicate (Fig. 2c). In addition to high-dimensional cell morphological features, we also included cell counts as an important measure to inform on cell stress, toxicity or proliferation. Heatmaps of cell counts (Fig. 2c) can reveal patterns among control wells that can serve as an indicator of both position and plate effects, and scatter plots of cell counts (Fig. 2d) can easily distinguish treatments with cytotoxic effects.

The value of using cell-level features, rather than simply well means or medians, is illustrated by examining the distribution of total DNA content, as measured by fluorescence intensity of the DNA stain Hoechst 33342 (Fig. 2e–g). Total DNA intensity is an indicator of cell cycle phase that is regularly measured in both flow cytometry[29] and HCS cell proliferation assays[17]. Under control conditions, this feature follows a bimodal distribution with peaks corresponding to 2n (G1 phase) and 4n (G2) genome content, which can only be detected by looking at the full distribution of DNA intensity (Fig. 2e). Comparisons of distributions between standardized treatments and controls can then be performed to detect defects in cell cycle transitions. For example, cells treated with mitoxantrone (an antineoplastic compound) elicited a dose-dependent phenotypic response, with a progressive shift in the ratio of the G1 and G2 peaks with increasing concentration (Fig. 2f). While the well medians of total nucleus intensity would show a shift in this case, well-averaged

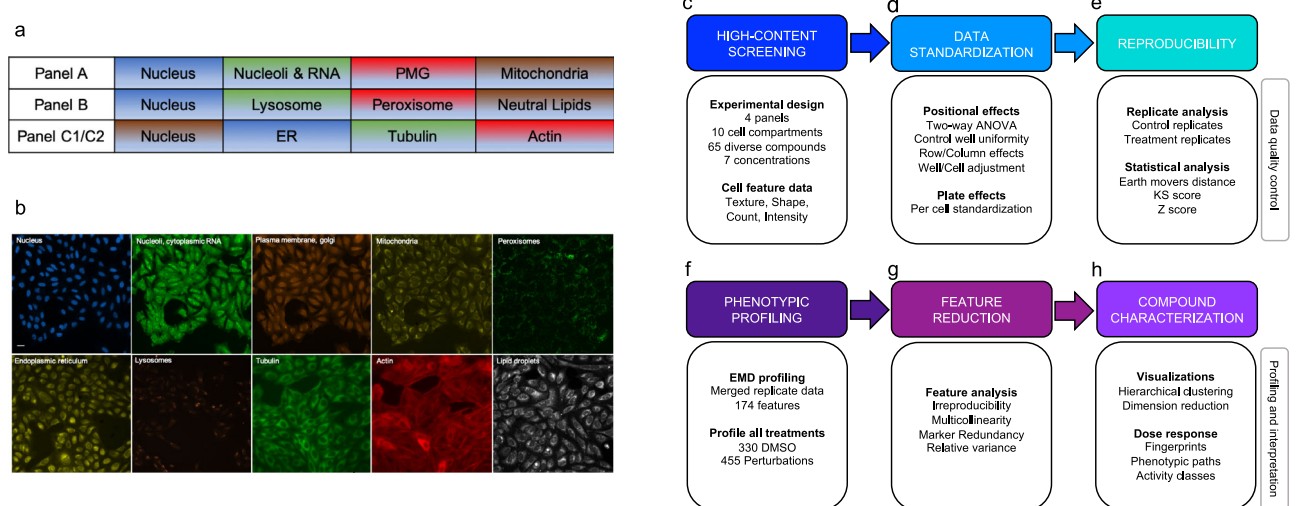

**Fig. 1 High-content screening (HCS) assay panels and data analysis workflow. a** The high-content screening (HCS) assay system comprises fluorescent stains and genetically encoded markers for ten cellular components, split across four panels. PMG: plasma membrane and Golgi, ER: endoplasmic reticulum. **b** Sample images of U2OS-labeled cellular components. Scale bar: 20 μm. **c-h** Overview of analysis workflow. **c** Assays are performed in 384-well plates. Cytological features capturing information on texture, shape, count, and intensity are measured for each panel using high-throughput microscopy. **d** Well- and cell-level data for each plate are adjusted and standardized for any positional (row or column) effects detected. **e** Assay reproducibility is assessed according to consistency among replicates of the same treatment. Three statistical metrics are compared by their sensitivity in detecting differences between replicates. **f** Consistent replicate data are aggregated and differences between treatments and controls are scored by comparing feature distributions. **g** Redundant, noisy, and uninformative features are removed. **h** Fully processed data are used to generate normalized phenotypic fingerprints, which are analyzed and visualized using hierarchical clustering and dimensional reduction.

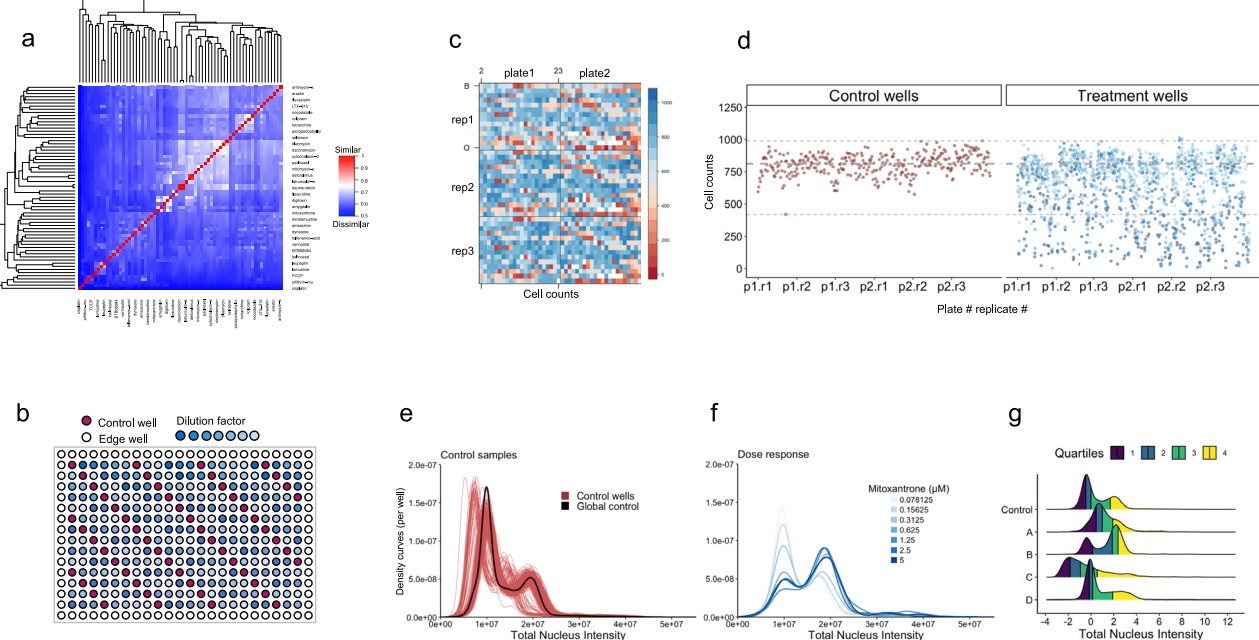

**Fig. 2 HCS experimental design and inspection of cell counts and cell cycle distributions. a** A set of 65 compounds with diverse chemical structures and a wide range of ChEMBL target classes were selected to examine the performance of the marker panels. Hierarchical clustering using the Tanimoto distance metric shows pairwise chemical structure similarity. **b** The 384-well plate layout for chemical screening: control wells (DMSO treatment) are placed in a diagonal pattern, and a dilution series of compounds arranged horizontally with high to low concentration (left to right). The outermost rows and columns are excluded from scanning and analysis to mitigate edge effects due to evaporation. **c** Dilution series of 65 compounds and control wells were spread over two plates, with three replicates for each condition. The heatmap shows well-level cell counts in different plates. **d** Scatter plot showing cell counts in control (left panel) and treatment (right panel) wells, ordered by plate and replicate number (grey dashed lines correspond to min, max and median of control cell count). **e** Hoechst 33342 (nuclear DNA) stain distribution showing density curves for individual control wells (red) and aggregated control wells (global control, black, sample size: 265,638 cells). **f** Distribution of total nucleus intensity of cells after the treatment with increasing doses of Mitoxantrone (sample sizes from low to high dosage ($n_{0.078125}$ – $n_5$): 711, 669, 559, 372, 325, 363, 363 cells). **g** Cells treated with different compounds and concentrations (A = irinotecan 5 μM, B = monensin 0.3125 μM, C = rapamycin 10 μM, D = vincristine 10 μM) show a diverse distribution of total nucleus intensity (sample size: Control = 265,638, $n_A$ = 1544, $n_B$ = 1817, $n_C$ = 1849, $n_D$ = 1173 cells).

data is unable to distinguish whether the observed response is due to a global shift in cellular feature distribution (Fig. 2g, sample B), a stretch of a distribution tail (Fig. 2g, samples C and D) or some other response (Fig. 2g, sample A). Therefore, we emphasize the use of cell feature distributions rather than well-averaged measures, since different treatments could lead to distinct subpopulations of cells with different characteristic responses.

Below we describe each component of the analysis workflow. We emphasize the importance of data preprocessing, describe statistical strategies for data integration and provide a comprehensive overview of methods for estimating robust fingerprints and broad-spectrum profiles across multiple staining panels and concentrations.

**Positional effects adjustment and data standardization**. A major issue when dealing with high-throughput data from technical replicates and different panels is distinguishing biological from technical variation, and most importantly recognizing meaningful treatment effects. Natural variability is inherent in multi-well-based assays and presents itself as random noise. In contrast, positional effects due to technical variability manifest as distinct spatial patterns across the rows, columns and edges in different plates, a common challenge in multi-well-based assays[30–33]. An important consideration in experimental design is the distribution of control wells across the plate. Placing controls in all rows and columns will reveal non-uniform positional effects that are easily detected by visual inspection of well-averaged heatmaps (Fig. 3a) which can be used to correct for technical artifacts. Our strategy was to automate the estimation of positional dependencies on each plate by applying a two-way ANOVA model for each individual feature on control wells (using well medians). Two-way ANOVA is suitable in this context since it examines the influence of two categorical variables (row and column position) on one numerical dependent variable (feature)[34].

We found that overall, fluorescence intensity features exhibit more positional effects than cell counts or morphological features such as cell shape (Fig. 3b). Almost half (45%) of all intensity-related features exhibited significant row or column dependency ($P < 0.0001$), whereas only 6% of morphological features such as spot, texture and shape, as well as cell counts, exhibited positional dependencies (Supplementary Fig. 1a, b). Row effects were detected more frequently (smaller $P$ values) than column effects, as seen in the ordered negative log of $P$-value plots (Supplementary Fig. 1a, b). This likely resulted from the way the automated liquid handler dispenses reagents (using a 12-well pipettor) and the sequence in which the HCS system scans 384-well plates row-wise along the wells. When comparing the performance of individual markers (Fig. 3b), intensity features derived from the RNA stain (Syto14) and DNA (DRAQ5 channel) showed the strongest positional dependency in all plates (Fig. 3b). Collectively, this approach allows us to efficiently and systematically assess the predisposition of different markers to positional effects in the early stages of the analysis phase.

When significant positional effects are detected among the control wells, the entire plate will be adjusted by the median polish algorithm[35], which utilizes the well medians to iteratively calculate row and column effects for each control and treatment well within each plate. Figure 3c shows an example of total nucleus intensity, where one plate (plate 1, replicate 1) exhibits clear row effects. The positional adjustment is displayed as the difference between the median polish adjusted output and the raw data. The B score, which is an analog of the Z-score, is then calculated by dividing the residuals within each plate by their median absolute deviation to account for plate-to-plate

changes[30]. This well-level adjustment and standardization yields harmonized and comparable replicate plates.

After adjusting for plate position effects, the data are further corrected at the cellular level to ensure that individual cell populations within each well reflect the newly adjusted well median by linearly scaling (adding or subtracting) the adjustment amount (Fig. 3d). To account for plate-to-plate variation, the cellular feature distributions are then standardizing to the control cells within each plate[36]. Each cell ($x_{ijk}$) is standardized by subtracting the median of control cells (numerator, Eq. (1)) and dividing by the MAD (median absolute deviation) of controls per plate (denominator, Eq. (1)), where letters ($i$, $j$, $k$) represent row, column, and plate respectively.

$$BZ_{ijk} = \frac{x_{ijk} - med\left(x_{control,k}\right)}{mad\left(x_{control,k}\right)} \qquad (1)$$

This two-level data normalization approach accounts for within-plate position effects and plate-to-plate technical variation, while also coercing cell feature distributions to follow a unitless score (which we call the per-cell BZ score, Eq. (1)). As demonstrated using both control cells (Supplementary Fig. 1c, i–k) and chemically perturbed cells (Supplementary Fig. 1d–h), different features inevitably exhibit positional and plate-to-plate variation, which without proper standardization would be carried through as unintended noise during downstream data aggregation. This preprocessing step further facilitates cell feature distribution comparisons when integrating datasets of multiple panels across plates and batches. Plate layout is an important design consideration for this step, as a poor plate layout (with inadequate numbers and positions of control wells) could hinder the proper identification of technical noise within a plate and lead to subsequent confounding of technical noise with true perturbations of biological signals.

**Statistical metric performance comparison using replicates**. All feature distributions for both treatment and control wells have been corrected and standardized across replicate plates based on per-plate control cell distributions (Fig. 4a). Using cellular data measured from 330 control wells and 455 chemical perturbations (×3 replicates), we show how this data can be interrogated to evaluate the performance of different statistical metrics for their ability to assess reproducibility among experimental replicates.

We tested the performance of three statistical metrics that can be used to detect differences between two feature distributions: the robust Z-score, the Kolmogorov–Smirnov (KS) test, and the Wasserstein distance. These rely on different characteristics of the cell feature distributions being compared, and each produces a different distribution of statistical scores across all features. The robust Z-score is sensitive to shifts in median, and commonly used as a normalization and strength of perturbation value in the context of image-based phenotypic profiling, but it has not been used for estimating replicate dissimilarity, nor compared to other distance metrics[12,30,37]. The KS test is a non-parametric test that measures the largest vertical distance between two empirical cumulative distribution functions (ECDFs) (Fig. 4b). The KS test detects shifts in location and shape between two CDFs based on a single measure of the maximal distance between them, and thus does not quantify overall differences between two sample distributions. The Wasserstein distance, also known as the earth mover's distance (EMD)[38], is a measure of the distance between two probability distributions on a given metric space. For univariate distributions where the metric space is $\mathbb{R}^1$, EMD can be approximated by the area obtained by integrating the

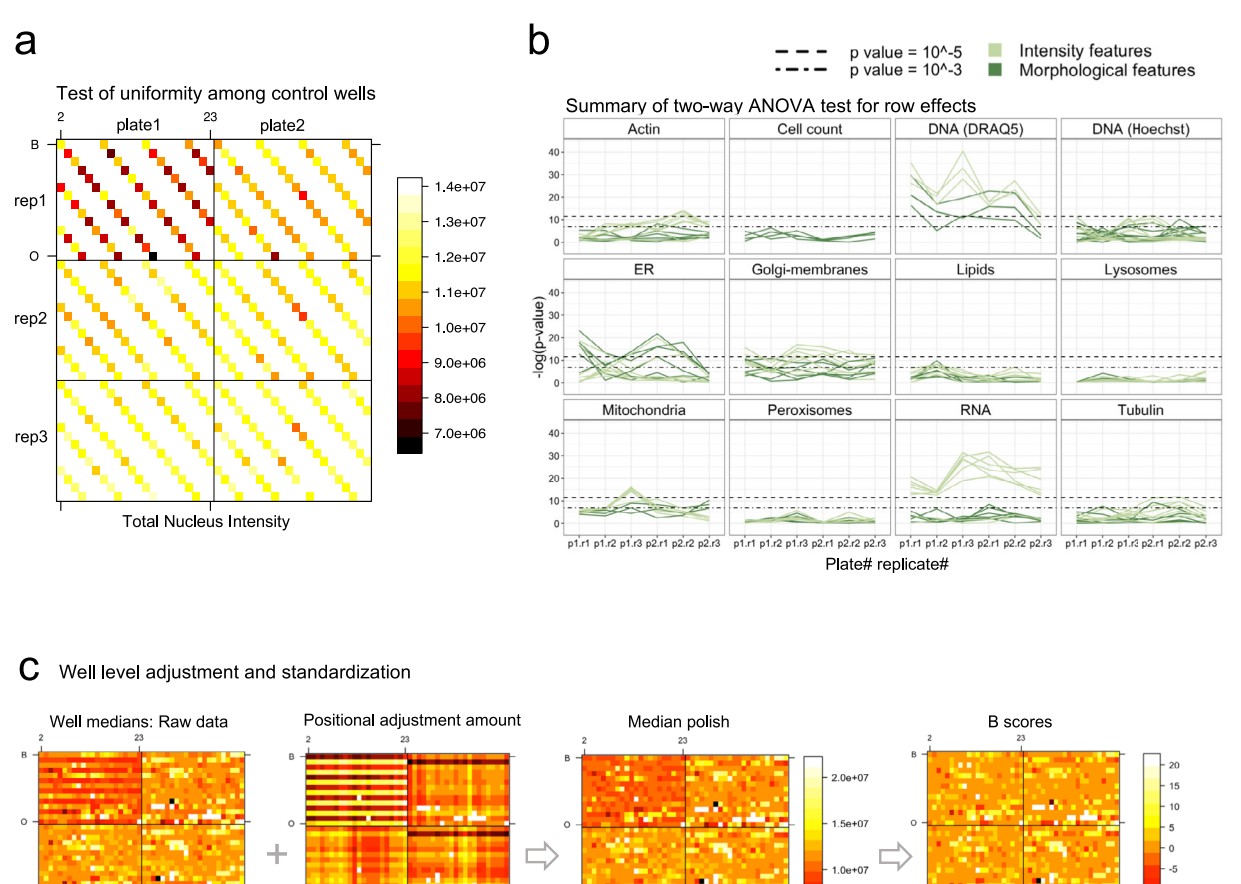

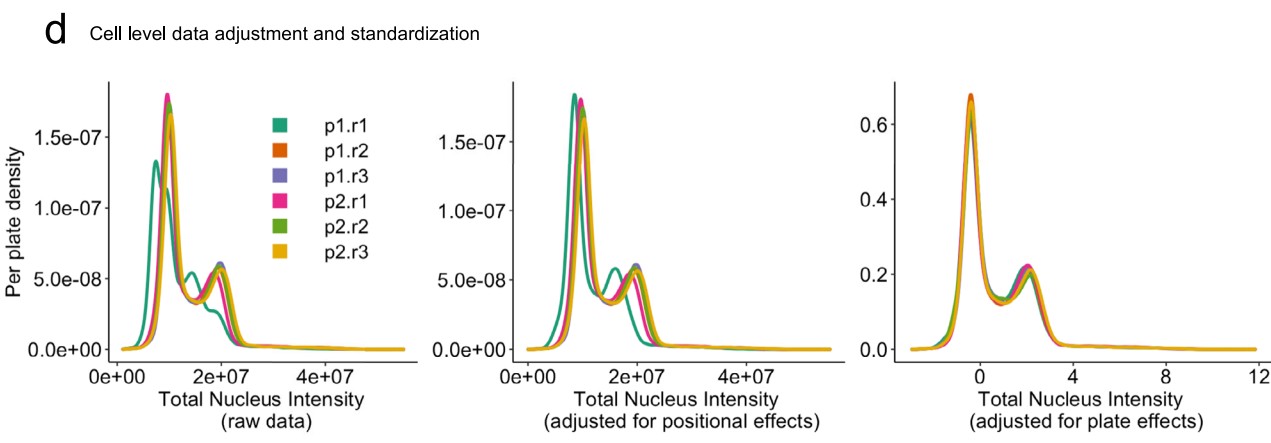

**Fig. 3 Adjustment of plate positional effects and data standardization across different plates. a** For each feature, a two-way ANOVA model is applied to detect non-uniformity among control wells due to row and column effects. Here total nucleus intensity is shown as an example, with the first replicate for plate 1 ("plate1 rep1") showing positional effects among control wells. All statistical output from two-way ANOVA analysis, including F-statistic and corresponding *P* values for both row and column effects detection, is provided in Supplementary Data 1. **b** Summary of row effects, shown as negative log of *P* values (*y* axis) for each feature across different plates (*x* axis), grouped by individual markers and cell counts. For example, the RNA marker shows a clear separation of intensity (light green curves) from non-intensity features (dark-green curves). **c** Well-level positional effects adjustment and standardization: when positional effects are detected (−log(*P*) >10) for a particular feature, median polish is applied. The B score standardizes well medians to per-plate controls to account for plate-to-plate variation. **d** Cell-level adjustment and standardization: cell populations in each well are adjusted for positional effects based on the adjustment amount calculated at well level. The adjusted cell-level data are further standardized to per-plate controls.

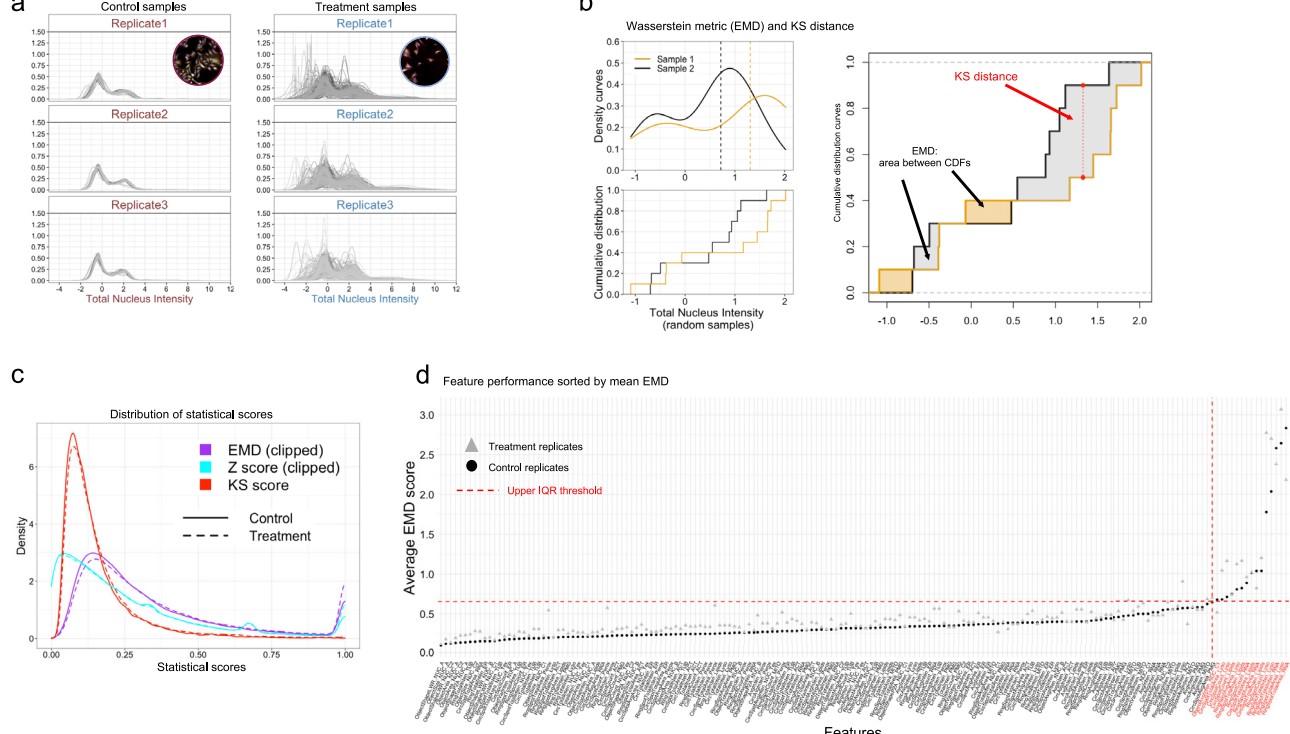

**Fig. 4 Statistical metric comparison and feature reproducibility. a** Feature reproducibility is assessed by estimating statistical distance among all pairwise replicates in both control samples (left) and treatment samples (right). **b** Hypothetical probability density (PDF) and cumulative density (CDF) curves for two random samples of the same feature are illustrated to show how the Kolmogorov–Smirnov (KS) distance and Wasserstein metric (EMD) are estimated. **c** Distributions of statistical scores measured by all pairwise differences between replicates are consistent among both treatments and controls, with EMD score showing higher sensitivity in detecting discrepancies. A full summary of replicate pairwise differences is provided in Supplementary Data 2 which lists feature, treatment (*compound_concentration*), plate, well id, sample size (as cell count), KS score, EMD score, and Z-score. **d** Features are sorted by their average EMD score between all replicates as an indicator of reproducibility. A high average EMD score indicates higher variation of a feature among replicates (low reproducibility). Features with poor reproducibility (outliers) falling above the upper interquartile (IQR) threshold value of 0.65 (upper threshold value = 1.5 × IQR + upper quartile) are highlighted in red.

absolute difference between two cumulative distribution functions (CDFs)[39] (described in Fig. 4b and Eq. (2)):

$$W_1(F_1, F_2) = \int_{-\infty}^{\infty} |F_1(x) - F_2(x)| dx \qquad (2)$$

The EMD score is unbounded and sensitive to differences in moments: shifts in mean, dispersion, skewness, and kurtosis. Hence, EMD will outperform both KS and Z-score metrics when distributions differ in one or more of these ways or show anomalies such as heavy tails, a common characteristic we observe in many cell feature distributions.

In order to assess the reproducibility of replicate assays, we used these three statistical metrics to measure the pairwise dissimilarity of individual feature distributions between replicate assays for both controls and treatment wells. We then visualized the distributions of the resulting scores among each of the 16 features measured for each marker using boxplots (Supplementary Fig. 2a–c). Although these metrics are bounded differently, as noted above, all three were able to distinguish the most stable features (e.g., DNA (Hoechst 33342)) from the noisiest ones (e.g., RNA (Syto14)) based on total variation overall and the presence of extreme outliers. However, while Z-scores did a better job separating these than the KS scores, both struggled to clearly distinguish the most variable features in comparison with EMD scores (Supplementary Fig. 2d–f). Therefore, EMD scores are better able to discriminate features with poor replication consistency than other metrics.

We also compared the reproducibility of replicate assays for controls versus treated samples to see whether they showed

similar levels of variability. We found that the distributions of pairwise differences between replicates were similar for all features in both datasets, regardless of the statistical metric used (Fig. 4c). Features sorted by the mean pairwise EMD among replicates revealed a subset of features that exhibited extreme variability, which appear as outliers in comparison with the upper IQR threshold (Fig. 4d, red line). Outliers primarily comprised features that measure the number of puncta ("spots") for the lysosome, lipid, and RNA markers.

Collectively, this analysis indicates that in our dataset, control and treatment samples showed similar levels of reproducibility among replicates, and that the EMD is an effective means to identify individual features that should be excluded from downstream analyses due to their low reproducibility even among the controls.

**Phenotypic profiling using the EMD score.** Since the common practice of using well averages or ensemble scores of replicate data is unable to inform on changes in the distributions of cell populations, we sought an alternative approach to exploit more of the phenotypic information in the HCS data. Above we illustrated how replicates can be used to investigate feature reproducibility (Fig. 4); however, some treatments result in reduced cell numbers, which limits the statistical power when comparing cell population distributions.

Here, we propose a more comprehensive approach, by merging replicate samples to form a larger cell population once the replicate data has been fully normalized. As an illustrative

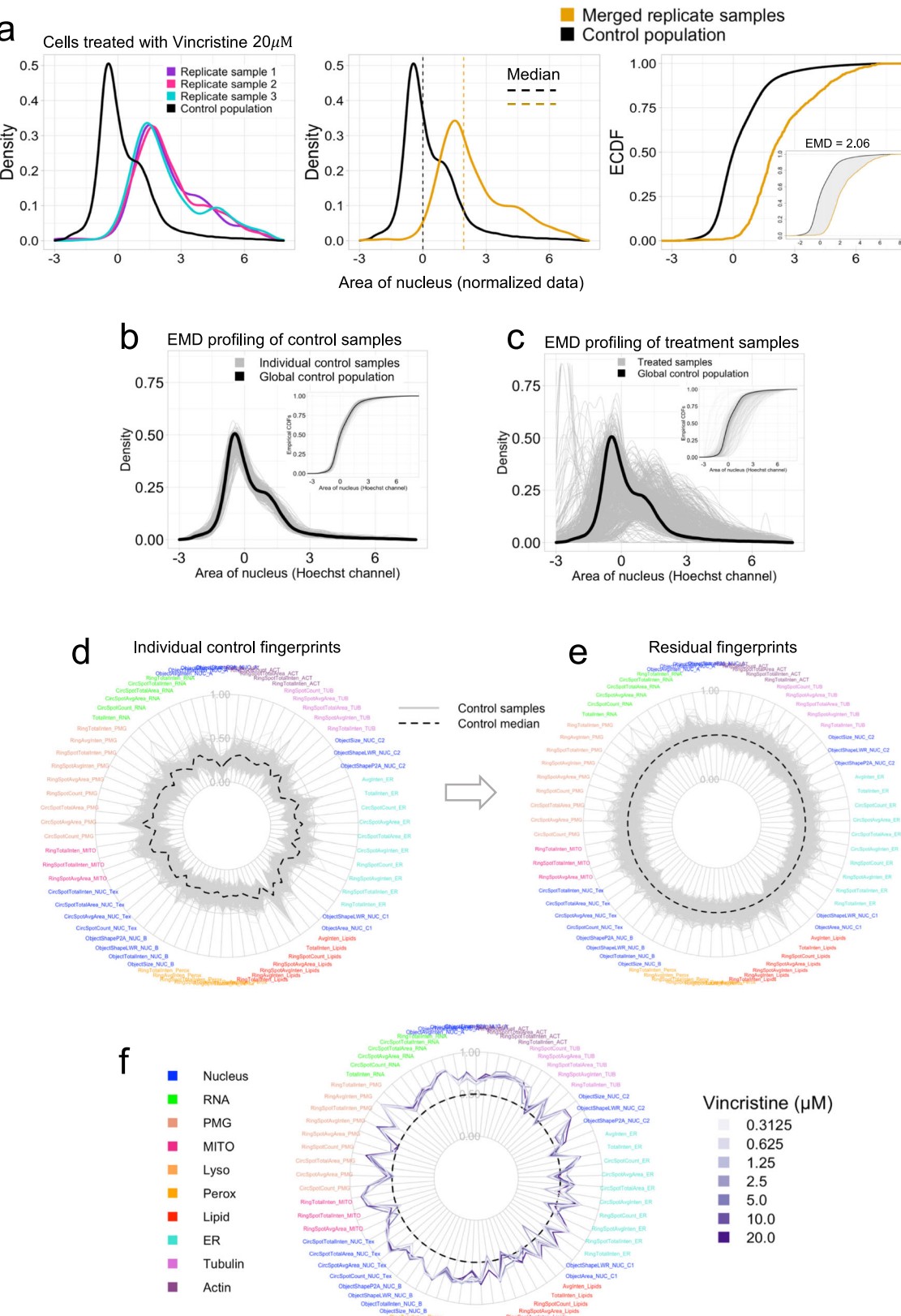

example, we use the area of the nucleus to establish the reference distribution for each feature (using DMSO controls) and combine all normalized control samples to form a global control population (Fig. 5a, black curve). Upon treatment with 20 μM Vincristine (a tubulin polymerization inhibitor), all normalized replicates show a consistently strong phenotypic response in this cell feature distribution compared to the global control distribution, indicative of an increase in nuclear area (Fig. 5a). This allows population data from replicate samples to be combined (Fig. 5a, orange curve, middle panel), from which the corresponding cumulative distributions (CDFs) can be generated for comparison (Fig. 5a, right panel).

**Fig. 5 Replicate aggregation and EMD profiling using global controls. a** Replicates of treatment samples with sufficient reproducibility are merged to form larger populations (orange curve) for subsequent EMD profiling relative to the global control (black curve). Shown is an example using the area of the nucleus feature with a strong phenotypic response to treatment with 20 µM vincristine (sample sizes: $n_{r1} = 348$, $n_{r2} = 386$, $n_{r3} = 366$, $n_{control} = 265,638$ cells). The empirical CDF (ECDF) curve shows a global right shift in the treatment condition, indicating an increase in global nuclear area. Inset: area between distributions measured by EMD (gray). **b, c** PDFs and (insets) CDFs for normalized control (per well) cell populations (**b**) and replicate-merged treatment populations (**c**). Differences between individual distributions relative to the global control are measured using the EMD metric and sample sizes corresponding to each statistical difference are listed in Supplementary Data 3 for 330 control ($n_{c-min} = 419$, $n_{c-max} = 990$ cells) and 455 treatment samples ($n_{t-min} = 52$, $n_{t-max} = 2763$ cells). **d** Radial plot of scaled EMD scores for 69 measured features among individual controls (gray lines). The EMD profile is log-transformed and min-max scaled to [0,1]. The median score of all controls fluctuates between 0.16 and 0.46 (black dashed line). **e** Radial plot of residual EMD scores of individual controls (gray lines) relative to the median score (black dashed line, zero). Residual score is defined as the difference between the score of the individual control and the median of all controls. The residuals naturally fluctuate around median zero with values between (−0.29, 0.45). The values have been offset by 0.5 to expand the plot for better visualization. **f** Radial plot of residual scores for Vincristine treatment at multiple concentrations, relative to control median. **d–f** Feature labels are color-coded by their corresponding cellular components.

As demonstrated in our analysis of reproducibility among replicates (Supplementary Fig. 2), since the EMD measures the full difference in mass between probability density functions, it has higher discriminatory power to detect differences between distributions of arbitrary shape in comparison to other statistical metrics. This principle can be applied to compare the global control population with individual control samples in order to assess overall technical variation after normalization (Fig. 5b). Of greater interest, we can use this metric to measure phenotypic responses of cell populations treated with different compounds at multiple doses (Fig. 5c).

After profiling all control and treatment samples for each feature, the full cytological profile can be summarized as a heatmap with treatment profiles sorted according to (per treatment) cell count (Supplementary Fig. 3). Incorporating cell counts in this way highlights the association between (increasing) strength of chemical perturbation (shades of blue in heatmap) and decreasing cell count. The profiles of all control samples are then used to generate a radial plot summarizing the variation among individual control samples for each of the ten marked cellular components, which we call a phenotypic "fingerprint" (Supplementary Fig. 4a; full-feature fingerprint and Fig. 5d; reduced feature fingerprint). For simplicity and ease of future comparisons, each sample fingerprint is subtracted from the control median to form new residual fingerprints (Supplementary Fig. 4b; full-feature fingerprint and Fig. 5e; reduced feature fingerprint). This approach preserves the variability within the control profiles and ensures all treatment profiles (per-compound and multiple concentrations) are thus standardized and visually comparable to the control. Similar plots can be used to summarize fingerprints using optimally reduced feature sets measured for individual compounds across the range of seven concentrations tested. For example, the anticancer therapeutic Vincristine elicited strong responses in features associated with its annotated cellular target, tubulin, as well as features for many other markers (Fig. 5f). Secondary phenotypes likely reflect indirect cellular responses to inhibition of tubulin polymerization, which also blocks mitosis and eventually leads to apoptosis. The phenotypic fingerprint did not change substantially with increasing concentration, suggesting that cells are maximally sensitive to even small doses of this compound.

**Hierarchical clustering and dimension reduction**. The reduced feature profile (described in "Identification of informative features and feature reduction") was then used for downstream exploratory data analysis and global comparison of phenotypic profiles. We first performed similarity analysis by hierarchical clustering using the set of 69 distinct features and found that the phenotypic profiles clearly separate the control and treatment groups (Fig. 6a). Clustering also revealed distinct groups of compounds that exhibit low levels of phenotypic activity overall (Cluster 2),

high activity toward specific features (Cluster 3), or a broader array of strong phenotypic responses (Cluster 4).

Visualization of the phenotypic profile in lower-dimensional space by uniform manifold approximation (UMAP)[40] similarly identified distinct clusters roughly corresponding to the broad classes identified above (Fig. 6a, b), and it additionally elucidated dose-dependent phenotypic patterns or "trajectories" across the different dimensions (Fig. 6b). The first and second UMAP dimensions separated the control group from the majority of treatment groups (Fig. 6b, inset). Plotting dimensions 2 and 3 further separated the controls from most of the Cluster 2 compounds, which we term the "low stress" cluster (Fig. 6b and Supplementary Fig. 5a–c). Treatments in this group show low toxicity, with no effect on cell counts and little overall effect on cytological phenotypes.

UMAP dimension 3 discriminated phenotypically active and toxic treatments from the low-activity and control groups and separates treatments in Clusters 3 and 4 from both controls and the low-activity group. Color coding each treatment by cell count (percent of control) indicates that heightened phenotypic response is associated with increasing toxicity (cell cycle arrest or cell death), as indicated by the decrease in cell counts from top to bottom (Fig. 6b). Cluster 3 treatments, which showed a range of specific phenotypic responses, tended to show intermediate effects on cell counts and segregate into smaller groups that are distributed across a wide range of coordinates in dimensions 2 and 3 (Supplementary Fig. 5g–i). Cluster 4 treatments (Fig. 6b, Toxic red zone; Supplementary Fig. 5j–l) were broadly active phenotypically and were cytotoxic. We refer to this as a "high stress" condition.

Thus, both hierarchical clustering and UMAPs provide a global overview of phenotypic profiles and display complementary information. While hierarchical clustering distinguishes broad phenotypic classes with low vs. high activity and specific vs. broad-spectrum cytological responses, UMAPs reveal treatment subgroups with distinct phenotypic responses and dosage-dependent phenotypic trajectories along a gradient of cytotoxicity.

**Phenotypic characterization of selected compounds**. After identifying broad phenotypic groups with global profiling methods, we next examined dose-dependent cellular responses, cell count and cell cycle distribution for representative compounds with different annotated mechanisms of action (Fig. 7a–c). Based on these criteria, each compound falls within one of the following activity groups: low stress, active (dose-insensitive), active (dose-responsive), and active (cytotoxic). We chose one representative compound with a distinct annotated MOA from each activity group to illustrate the major differences between the groups.

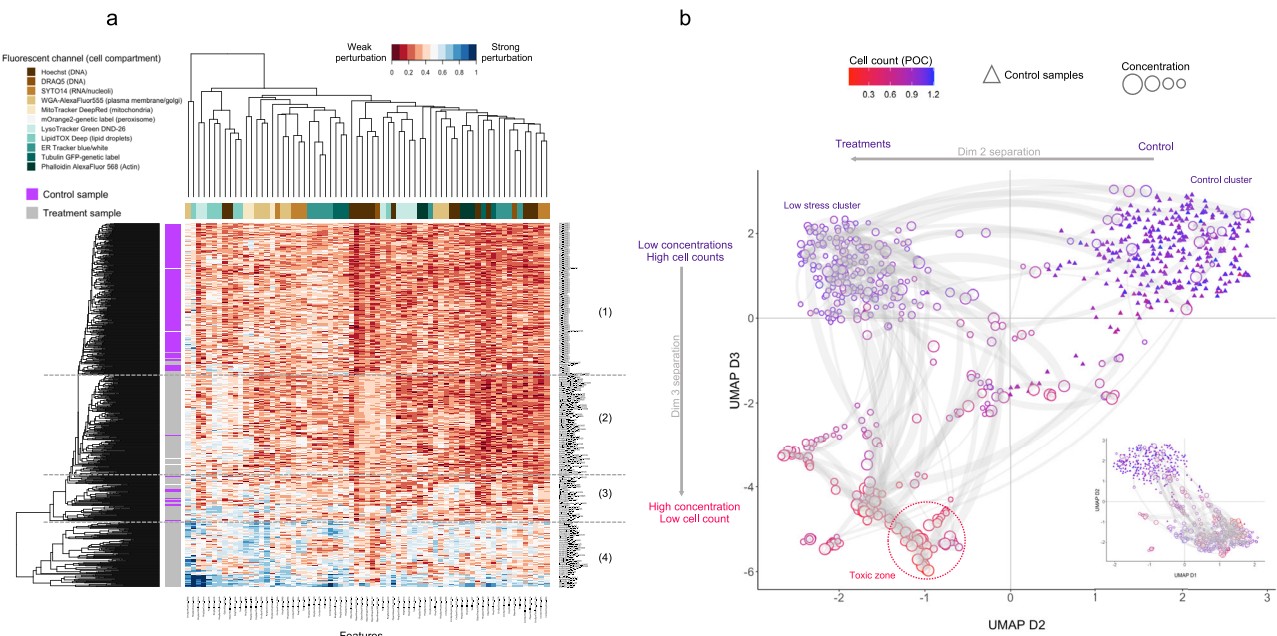

**Fig. 6 Hierarchical clustering and dimension reduction. a** Hierarchical clustering of EMD scores for control and treatment profiles shows four main clusters: (1) control cluster, (2) "low stress" cluster, (3) active phenotypes, (4) broad phenotypic responses with cytotoxicity. **b** UMAP dimensional reduction of phenotypic profiles based on EMD scores for all samples. Dimensions 1 and 2 separate most controls (triangles) from treatment samples; dimension 3 additionally separates treatments by cell count. Light gray curves represent per-compound phenotypic trajectories along a concentration gradient.

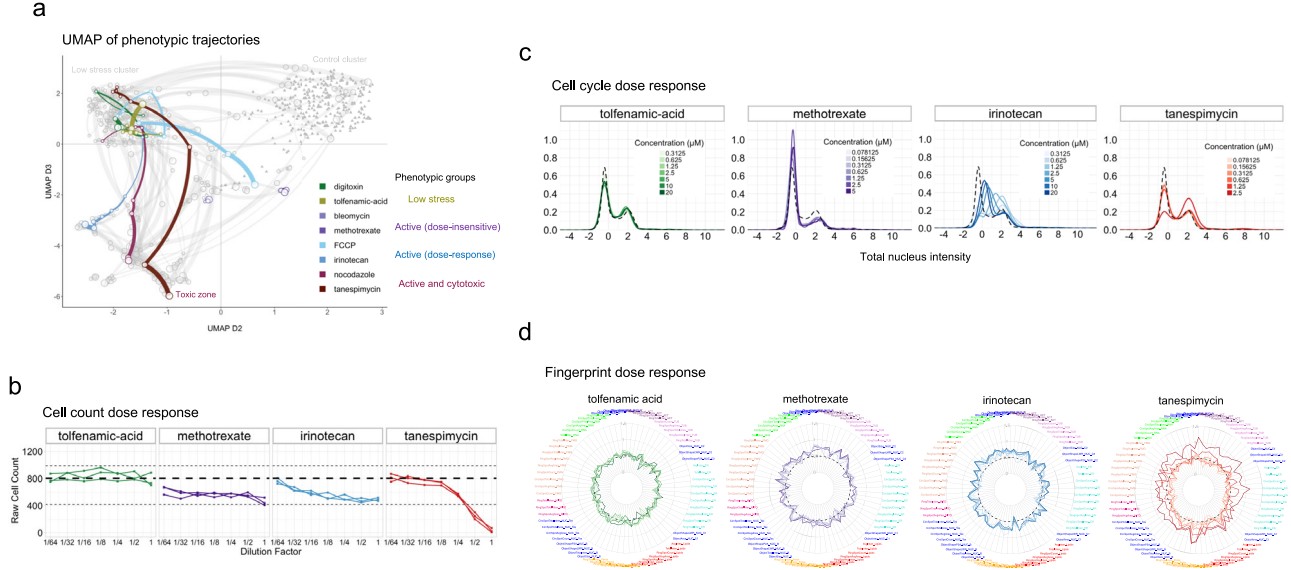

**Fig. 7 Phenotypic characterization of selected compounds. a** Phenotypic trajectories of selected compounds from different activity groups are highlighted in the UMAP. Green: low stress; purple: active (dose-insensitive); blue: active (dose-responsive); red: active and toxic. **b** Raw cell counts across the dilution series of representative compounds in each activity group. Each curve indicates a replicate; the dashed line represents the mean, and dotted lines the maximum and minimum of control cell counts. **c** Cell cycle distributions (total nucleus intensity) across the compound dilution series for four representative compounds from each activity group, color-coded as in (**a**). The full set of compounds as described in (**a**–**c**) are provided in Supplementary Fig. 5. **d** Radial plot of fingerprints for representative compounds from (**c**) at each of the seven concentrations measured. Feature labels are color-coded by their cytoplasmic component.

*Low stress.* Tolfenamic acid (TA) elicits a minimal phenotypic response; its phenotypic UMAP path shows little movement within the low-activity cluster, and it affects neither cell counts nor cell cycle distribution (Fig. 7a–c). TA is an inhibitor of the enzyme cyclooxygenase (COX), also called prostaglandin-endoperoxide synthase (PTGS), which is involved in the conversion of fatty acids to prostaglandin[41] and is targeted by a variety of anti-inflammatory drugs[42]. While TA is reported to exert anticancer activity in medulloblastoma[43], colon cancer[44], and head and neck cancer[45], there are no reports on the effects of TA on U2OS cells. Since COX/PTGS enzymes are typically induced in response to inflammation in vivo[46] and are not

found to be expressed in U2OS cells according to the Human Protein Atlas database (https://www.proteinatlas.org/ENSG00000095303-PTGS1/cell+line), this compound is not expected to show strong and specific phenotypic responses in this cell model (Fig. 7d).

*Active (dose-insensitive)*. Methotrexate (MTX) is a chemotherapeutic agent that inhibits DNA synthesis by targeting dihydrofolate reductase[47], an enzyme needed for biosynthesis of nucleic acid precursors and some amino acids. MTX elicits strong phenotypic effects that are relatively consistent across all seven doses (Fig. 7a). MTX reduces cell counts by ~25% and induces a G1 arrest phenotype, as revealed by the increased proportion of cells in G1 phase and corresponding decrease in G2/M (Fig. 7b, c). This observation is consistent with its reported inhibition of DNA synthesis during S phase[48]. The radial plot of the phenotypic fingerprint shows major responses in nuclear, tubulin, and actin features (Fig. 7d), as may be expected in response to a cell cycle blocker.

*Active (dose-responsive)*. Irinotecan (IRI) is a chemotherapeutic agent that inhibits topoisomerase I activity, which in turn inhibits both DNA replication and transcription[49]. IRI shows a pronounced phenotypic dose response: its UMAP trajectory begins in the low-stress region (top left) and travels downward along dimension 3 (Fig. 7a, b). This reflects a progressive decrease in cell counts with increasing concentration, although the compound does not cause severe cytotoxicity at any of the concentrations tested. Notably, cell cycle phenotypes differed in a dose-dependent manner: at low concentrations IRI induced a G2/M block, which shifted toward a block at G1/S with increasing concentration (Fig. 7c). A previous study also reported a significant increase in cells at S and G2/M in human colorectal cell lines upon IRI treatment[50]. Phenotypic fingerprints also showed dose-dependent changes in actively responding cytological features (Fig. 7d).

*Active (cytotoxic)*. Tanespimycin binds to and inhibits heat shock protein 90 (HSP90)[51] and is known to be toxic in higher doses (Fig. 7b). Its UMAP trajectory starts in the low-stress cluster, but transitions to the cytotoxic zone at higher concentrations (Fig. 7a). Lower concentrations show cell cycle distributions similar to controls, with an abrupt G2/M arrest at 2.5 μM (Fig. 7c). The phenotypic fingerprint displays a strong dose-dependent response in a wide range of features, with extreme phenotypic changes at the highest concentration due to cytotoxicity (Fig. 7d).

In summary, these examples highlight the benefits of incorporating cell cycle, cell counts and dose responsiveness in characterizing compound activity. Cell count is one of the simplest and easiest measurements to interpret, as it reveals the level of cytotoxicity induced by a chemical treatment perturbation. Cell cycle distributions are also highly informative, as many compounds interfere with cell cycle progression through different routes. Examining activity levels of each compound across a concentration range adds another layer of information for distinguishing treatment profiles, for instance IRI and Nocodazole (NOC) appear phenotypically similar at low concentrations but then diverge at high concentrations (Fig. 7a). Their similarity at lower concentrations could be due to their mild response to treatment, which we observe in their phenotypic fingerprints, biological images, and cell feature distributions (Supplementary Fig. 6a, c–g). Each fingerprint at their highest concentration, however, induces increased activity (with larger discrepancies between the two compounds) in several distinct feature channels (Supplementary Fig. 6b). The overall trajectories reflect the different degree of dose-dependent effects of these two compounds, which may be due to differing MOAs (IRI directly targeting DNA processing via inhibition of topoisomerase I vs. NOC interfering with microtubules).

## Discussion

Since inferences drawn from raw measurements of cytological features largely depend on how samples are prepared and how experimental data are collected, processed and reported, developing robust strategies for data collection and analysis are key. However, despite ongoing collaborative efforts, to date the HCS community has not yet converged on a set of standard best practices for handling such issues at any stage of the analysis. Here, we present a high-content screening assay based on a comprehensive set of cytological features, together with a robust statistical analysis workflow, to profile broad-based cellular phenotypic responses to small molecules or genetic perturbations. The workflow performs quality control and preprocessing of image-based data, feature reduction, generation of phenotypic fingerprints, and visualization of phenotypic responses. Our combined experimental platform and analysis framework introduces and outlines strategies to address a number of important issues in HCS data collection, processing and analysis of high-content cytological phenotypes.

First, positional effects (edge effects, row/column dependencies, or gradient artifacts) are a persistent factor in multi-well assays and microarray experiments[30,31,52]. Ideally, to mitigate such effects samples should be randomly placed within the plate of different replicates. However, this is impractical for high-content screening projects with hundreds to thousands of compounds. To control for positional artifacts, our strategy was to design a 384-well plate layout with 55 control wells placed in a diagonal pattern, so that each row and column has a sufficient number of control samples. We demonstrated the benefits of spreading out the controls as an alternative to the more common practice of confining control samples to certain columns[53] by showing that this plate configuration can feasibly capture problematic spatial patterns, in particular prominent row and column position dependencies.

Second, although cellular features exhibit diverse distributions, this information is rarely exploited in the analysis of HCS data, which instead relies primarily on well-averaged data. We hypothesized that quantifying feature variability in control populations can both provide vital information at the quality control stage and serve as a key element in each step of the data processing workflow. The analysis framework we developed demonstrates the benefits of incorporating variability among control wells as a strategy to assess replicate reproducibility by comparing cell populations among control and treatment replicates. Moreover, incorporating variation of per-cell feature data has distinct advantages for feature reduction and downstream phenotypic profiling, which are essential for interpreting cytological responses to cellular perturbations.

One application of using feature distributions is to compare the performance of different statistical metrics in detecting differences between populations of cells. The Z-score relies on averaged well values and is sensitive to shifts in central tendency, whereas the KS test and EMD measure statistical distances between cell feature distributions based on maximum vertical distance and total difference between empirical cumulative distribution functions (ECDFs), respectively. Using these measures to compare experimental replicates, we showed that EMD exhibits higher sensitivity due to its ability to account for the area between two ECDFs, which captures arbitrary differences in distribution shape; in contrast, KS measures only a maximal difference in height between CDFs and is insensitive to multimodal

distributions. The EMD was originally conceived as a solution to the transport problem from linear optimization[54] but is regularly applied in different fields including image processing, pattern recognition (text processing), machine learning and flow cytometry data[55]. Although the EMD offers advantages in detecting a wide range of responses, one of its limitations is that it does not take directional changes into account. Extending this method to account for other forms of variation, including direction, could further improve the performance of this metric in downstream profile similarity analysis.

Screening and profiling of 65 compounds with diverse chemical structures and reported MOAs revealed broad concentration-dependent patterns of global cellular responses to chemical challenge. By combining cell count information with dose-dependence of phenotypic responses, four major treatment groups could be distinguished that we interpret as reflecting the level of stress imposed by different chemical perturbations. The "low stress" group showed minimal changes in both counts and phenotypic profiles in comparison with controls. Two classes with more pronounced phenotypic responses were associated with moderate reductions in cell counts, indicating an escalating level of stress on the cells. These phenotypically "active" groups differed in sensitivity to increasing compound dosage, showing either no changes in responsiveness or a gradient of responses that correlated with decreases in cell counts, which we interpret as reflecting increasing levels of cell stress. A fourth group showed strong cytotoxicity at one or more concentrations, reflected by broad phenotypic changes and dramatic reduction in cell counts.

A major goal of HCS studies is to identify compounds with similar MOAs based on phenotypic profiling. Because we chose a diverse group of compounds with different annotated MOAs that show little structural similarity, this particular dataset is not ideal for compound similarity or classification based on shared MOA. However, our observation that compounds with different MOAs cluster together at some concentrations but not others suggests that there is no straightforward way to perform mechanistic annotation based on phenotypic profiles at single concentrations. Most published HCS studies do not measure how cellular responses change as a function of compound dosage, nor do they treat cell count as an indicator of stress response[14]. While screening at multiple concentrations incurs additional experimental complexity and investment of time and resources, we found that dose-dependent phenotypic trajectories can provide additional layers of information that assist in discriminating the activity of different compounds, reinforcing observations from a previous study that sought to classify compound MOAs based on dose-dependent trajectories[21]. Thus, we believe that the concentration-dependent phenotypic trajectories revealed in the UMAPs (Supplementary Fig. 7a, g, j, l) hold promise for mechanistic discrimination, warranting a fuller characterization in future studies.

In summary, HCS is an emerging field that is still evolving rapidly in terms of experimental implementations and analytical approaches. In addition, HCS is used to address many questions in many different biological systems. Hence, the community has not developed widely accepted common standards for experimental and analytical workflows. These factors limit the ability to compare data from different studies, as well as the potential for data integration, which has proven to be very powerful in other domains such as genome-wide molecular profiling. The goal of our study was to contribute new methods that can help advance developments in this field. First, the novel HCS screening platform we introduce here offers a more comprehensive toolbox for surveying cytological responses to chemical or genetic perturbations by allowing the simultaneous measurement of phenotypic features for ten cellular compartments and components. Applying this expanded toolbox to screen diverse compounds at multiple concentrations, we also developed a new statistical framework and workflow for automated quality control, improved data standardization and phenotypic profiling that exploits the variation in phenotypic feature distributions, a hitherto underutilized source of information on cytological phenotypes. We believe that these innovations offer useful contributions to the field, and we hope that they may spark further interest and methodological developments that may facilitate the standardization and harmonization of HCS data from different labs.

## Methods

**Compound selection.** All chemical compounds used in this study are from the Selleckchem Bioactive Compound library (Cat. number: Catalog No. L1700) and were selected to represent diverse MOAs and effects on different cellular targets. Detailed information for each compound and the dilution series are provided in Supplementary Data 4, including information on MOAs and/or known biological targets, as well as specific functional annotations (e.g., topoisomerase, mitochondrial enzymes, HDAC inhibitor).

**Cell lines and cell culture.** U-2 OS (ATCC® HTB-96™), U-2 OS-mOrange2-Peroxisome and U2OS-LMNB1-TUBA1B-ACTB (Sigma Aldrich, Cat. number: CLL1218) cell lines were cultured using McCoy's 5A medium (Sigma Aldrich, Cat. number: M9309) supplemented with 10% fetal bovine serum (FBS; Thermo Fisher Scientific, Cat. number: 10082147) and 100 units of penicillin–streptomycin solution (Sigma Aldrich, Cat. number: P0781), in a humidified incubator at 37 °C with 5% $CO_2$. U2OS-LMNB1-TUBA1B-ACTB are derived from the parental U-2 OS cell line (ATCC® HTB-96™) and were genetically modified to contain three distinct fluorescently tagged proteins expressed from their endogenous loci: BFP-LMNB1, GFP-TUBA1B and RFP ACTB.

To genetically label peroxisomes, $2 \times 10^6$ parental U-2 OS cells (ATCC® HTB-96™) were seeded into two wells of a six-well plate and cultured under standard conditions for 24 h before transfection with 1 µg of the mOrange2-Peroxisomes2 plasmid (Addgene, Cat. number: 54596) using X-tremeGENE™ HP DNA transfection reagent (Sigma Aldrich, Cat. number: 6366244001) according to the manufacturer's instructions. After 24 h the transfection medium was replaced by fresh cell culture medium, and cells were cultured for another 24 h. Subsequently, transfected cells were seeded into 96-well plates at a density of ten cells per well and selected for stably transfected cells using 300 µg/ml G418 (Sigma Aldrich, Cat. number: A1720) for 2–3 weeks. Emerging cell clones were transferred into 24-well plates and expanded until sufficient cells were yielded to prepare cryo stocks for U-2 OS-mOrange2-Peroxisome cells.

**Cell seeding for HCS experiments.** All U-2 OS cells used in this study were grown in T75 or T185 cell culture flasks (Thermo Fisher Scientific) until confluency reached 70–80%. Cells were harvested by TrypLE (Thermo Fisher Scientific, Cat. number: 12604013) and cell numbers were determined using EVE™ Automated Cell Counter (NanoEnTek). Cells were seeded into 384-well plates (Greiner Bio-One black µClear®, Cat. number: 781091) at a density of 1800 cells per well in 32 µl of McCoy's 5 A medium supplemented with 10% FBS and 100 unit of penicillin–streptomycin using a Matrix WellMate liquid handling device (Thermo Fisher Scientific) placed in a laminar flow hood. After seeding, plates were kept at room temperature for 30 min and then transferred to an incubator with a rotating plate hotel (Cytomat, Thermo Fisher Scientific). Compound treatment started 24 h after cell seeding.

**Compound treatment and plate layout.** The source plates containing serial dilutions of the compound and DMSO controls were prepared by combining McCoy's 5A medium (no FBS added) and various concentrations of test compounds in 384-well plates (Corning, Cat. number: CLS3657), with a specific diagonal pattern of controls (DMSO only). This configuration places multiple non-adjacent control wells in each row and column, which allows for the identification of plate positional effects. The source plates containing controls and serial dilutions of compounds were prepared, sealed by aluminum foil, and spun briefly to collect the solutions on the bottom of each well. Twenty-four hours after cell seeding, 8 µl of compound dilutions and controls from control plates were added to each well of replicate cell culture assay plates using a Bravo automated liquid handling platform (Agilent) at a final maximum DMSO concentration of 1%. After compound addition, plates were centrifuged for 1 min at 500 rpm and transferred to a Cytomat incubator. Cells were subject to each treatment for 24 h before staining.

**HCS staining panels.** Four different cell-staining protocols ("panels") were applied to three sets of assay plates. The details of cytological markers, their cellular targets, spectral properties, and suppliers are described in Table 1. All buffers and staining reagents were added using a Matrix WellMate liquid handling device (Thermo Fisher Scientific). After the addition of each reagent, plates were briefly spun in a

**Table 1 Cellular markers used in HCS panels.**

| Cellular target | Dye/marker | Excitation max[a] | Emission max[a] | Supplier/Cat. number |
|---|---|---|---|---|
| Panel A—cell line: Parental U-2 OS cells | | | | |
| Nucleus | Hoechst 33342 | 350 | 453 | Thermo Fisher Scientific/62249 |
| RNA and nucleoli | SYTO14 Green | 505 | 524 | Thermo Fisher Scientific/S7576 |
| Plasma membranes and Golgi | WGA Alexa Fluor® 555 Conjugate | 555 | 568 | Thermo Fisher Scientific/W32464 |
| Mitochondria | MitoTracker® DeepRed FM | 641 | 662 | Thermo Fisher Scientific/M22426 |
| Panel B—cell line: U2OS-mOrange2-Peroxisome cells | | | | |
| Nucleus | Hoechst 33342 | 350 | 453 | Thermo Fisher Scientific/62249 |
| Lysosomes | LysoTracker Green DND-26 | 500 | 510 | Thermo Fisher Scientific/L7526 |
| Peroxisomes | mOrange2 (genetically encoded) | 548 | 562 | Addgene/54596 |
| Lipid droplet | LipidTOX™ HCS LipidTOX DeepRed | 634 | 652 | Thermo Fisher Scientific/H34477 |
| Panel C1—cell line: U2OS-LMNB1-TUBA1B-ACTB cells | | | | |
| Nucleus | DRAQ5 | 596 | 696 | Thermo Fisher Scientific/62251 |
| ER | ER-Tracker™ Blue-White DPX | 371 | 557 | Thermo Fisher Scientific/E12353 |
| Panel C2—cell line: U2OS-LMNB1-TUBA1B-ACTB cells | | | | |
| Nucleus | Hoechst 33342 | 350 | 453 | Thermo Fisher Scientific/62249 |
| Actin | Alexa Fluor™ 568 Phalloidin | 578 | 603 | Thermo Fisher Scientific/A12380 |
| Tubulin | TUBA1B-GFP (genetically encoded) | 489 | 509 | NA |

[a]According to Fluorescence Spectra Viewer (Thermo Fisher Scientific).

centrifuge to collect liquids at the bottom of the wells. PBS, fixation solution (paraformaldehyde) and permeabilization solution (Triton X-100) were freshly prepared and filtered by a 0.2-μm membrane prior to use.

**Panel A (nucleus-RNA/nucleoli-endomembrane system-mitochondria).** Parental U-2 OS cells were incubated with a solution of mitochondrial dye (0.22 μg/ml; MitoTracker® DeepRed FM, Thermo Fisher Scientific) in cell culture medium (including 2% FBS) for 35 min under standard cell culture conditions. For fixation, the staining solution was removed and cells were incubated with 25 μl of para-formaldehyde solution per well (4%; Sigma Aldrich) for 20 min at room temperature. After removal of fixing solution cells were washed with 60 μl of filtered PBS and permeabilized with a 0.1 % Triton X-100 solution (freshly prepared in PBS; 25 μl per well) for 15 min at room temperature. Subsequently, cells were washed three times with 60 μl of PBS and stained with Hoechst 33342 (1:10,000 dilution; Thermo Fisher Scientific) and wheat germ agglutinin (7.5 μg/ml; Wheat Germ Agglutinin, Alexa Fluor® 555 Conjugate, Life Technologies, Thermo Fisher Scientific) for 45 min at room temperature and protected from light. After one washing step with 60 μl PBS a SYTO14 staining solution (2.5 μM; Thermo Fisher Scientific) was added, plates were sealed, incubated for 30 min at room temperature, and finally transferred to a fridge for storage until image acquisition.

**Panel B (nucleus-lysosomes-peroxisomes-lipids).** U-2 OS-mOrange2-Peroxisome cells were incubated with lysosomal dye (0.1 μM; LysoTracker Green DND-26, Thermo Fisher Scientific) in pre-warmed cell culture medium for 35 min under standard conditions. The staining solution was removed and cells were fixed with 4% paraformaldehyde for 20 min at room temperature. After one washing step with 60 μl of PBS cells were stained using 14 μl of the lipid staining reagent (1:750 dilution of stock solution, LipidTOX™ HCS LipidTOX DeepRed, Thermo Fisher Scientific) per well for 45 min. Finally, 25 μl of Hoechst 33342 nuclear staining solution per well was added on top, and after incubation, for 30 min at room temperature, the staining solution was removed and replaced by 70 μl of PBS. Plates were sealed and transferred to a fridge for at least 6 h prior to imaging.

**Panels C1 (nucleus–ER).** U2OS-LMNB1-TUBA1B-ACTB cells were incubated with an ER staining solution (1:1000 v/v dilution of stock solution, ER-Tracker™ Blue-White DPX, Thermo Fisher Scientific) in cell culture medium (including 2% FBS) for 35 min under standard cell culture conditions. Next, the staining solution was removed and cells were incubated with 25 μl of paraformaldehyde solution per well (4%; Sigma Aldrich) for 20 min at room temperature. After a washing step with 60 μl of PBS, cells were stained using 16 μl of the nuclear staining reagent DRAQ5 (2.5 μM, DRAQ5™ Fluorescent Probe Solution, Thermo Fisher Scientific) per well for 60 min. After removal of the staining solution 60 μl of PBS were added per well, plates were sealed and kept in a fridge until imaging.

**Panel C2 (nucleus–tubulin–actin).** After the image acquisition step of U2OS-LMNB1-TUBA1B-ACTB cells in panel C1, the solution in the plate was removed. Cells were then re-stained with 25 μl of a solution prepared from Hoechst 33342 (1:10,000 dilution; Thermo Fisher Scientific) and phalloidin (Alexa Fluor™ 568 Phalloidin, Thermo Fisher Scientific) for 45 min at room temperature. Subsequently, cells were washed twice with 60 μl of PBS, plates were sealed and transferred to a fridge until the 2nd image acquisition step.

**Image acquisition and data extraction.** Images were acquired using the Cellomics ArrayScan XTI platform (Thermo Fisher Scientific) equipped with a ×20 objective (Zeiss Plan Neofluar, NA 0.3) and an LED light source for wide-field fluorescence imaging. Fixed time exposure mode was used for each channel, and the exposure time was experimentally determined at less than 30% pixel saturation. A total of 9 fields in each well of the 308 inner wells of the 384-well plate were imaged.

Image analysis was performed using the Compartmental Analysis Bio Application package in the Cellomics software (Thermo Fisher Scientific). The nuclei with Hoechst 33342/DRAQ5-staining were identified as primary objects (Circ), and a simulated cytoplasm (Ring) was created according to nuclear shape and neighboring cells. The compartment analysis performs fluorescent quantification in both the nuclear (Circ) and the cytoplasmic (Ring) region of each valid cell. A total of 174 texture, shape, count and intensity features across all four panels were extracted with Cellomics software, which are listed in Supplementary Data 5 ("full feature set").

**Identification of informative features and feature reduction.** In our broad-spectrum assay, we cast a wide net to expand and diversify the feature space. Since high-dimensional datasets often contain some correlation structure, the number of cell features is routinely reduced in order to identify uninformative features, avoid redundancy and lower dimensionality for classification, visualization and interpretation. To understand which marker features produce biologically informative and non-redundant phenotypic signatures, we sought to eliminate irreproducible, highly correlated and low-activity features, as well as those deemed to have little biological relevance. Below we describe each of these steps, their rationale, and specific examples.

*Irreproducible features.* First, we identified a set of 15 irreproducible features based on their dissimilarity across replicates (Fig. 4d and Supplementary Data 5, "irreproducible"). As noted previously (described in "Statistical metric performance comparison using replicates"), many of the features measured for the lysosomal, lipid and RNA markers tend to have high variation among controls and replicates. Most of these tended to have questionable biological significance based on the measurement type and location within the cell. For example, since lipid droplet and lysosomal staining should be measured in the cytoplasm, nuclear signals for these markers most likely represent background noise.

*Biologically irrelevant features.* All features for each marker are measured within both the Nucleus (Circ) area and Cytoplasmic (Ring) area of the cell. However, features measured in the cytoplasm are not expected to be meaningful for nuclear markers (e.g., DNA), and likewise nuclear features are not expected to be meaningful for markers of cytoplasmic structures (e.g., lysosome, peroxisome, lipid droplet and tubulin). A total of 30 features were therefore removed by this filter (Supplementary Data 5, "circ features").

*Redundant features.* With high-dimensional data, it is desirable to reduce the feature set by removing uninformative signals that contribute little additional information. This can be done either by removing redundant features or using dimensional reduction methods such as principal components analysis (PCA)[13]. To preserve interpretability, we chose to first remove features with weaker or noisier signals that are largely overlapping with stronger, more robust signals. For example, lysosomal signals were very weak compared to those from the

peroxisomal marker, which is genetically encoded. Since these two compartments are labeled in the same panel and their fluorescence emission spectra overlap to some degree, weak signals in the lysosome channel that overlap peroxisomal features tended to represent bleed-through from the peroxisome channel (Supplementary Fig. 7a). Since lysosome and peroxisome features were highly correlated, we chose to remove the weaker lysosome features (Supplementary Data 5, "lyso features").

However, due to the important roles of lysosomes in the degradation and recycling of cellular waste, cellular signaling and energy metabolism, we remain interested to learn more about the activity of chemical compounds on lysosomes and to integrate this information with activities on other cellular markers. In future studies a genetically encoded lysosomal marker (e.g., mKO2-LAMP1) or a different chemical fluorophore with a stronger signal could replace the lysosomal marker used in this study.

The remaining feature set (124 features) was further filtered by removing weaker features from pairs of features with a Pearson correlation coefficient of 0.9 or above (Supplementary Fig. 7b and Supplementary Data 5, "correlated features"). Using these criteria, a total of 37 features were removed.

*Relative feature variance*. Per-feature variance can indicate responsiveness to chemical perturbations. Zero or low variance of a feature across the full range of treatments suggests that it is relatively insensitive to perturbations and thus of little value for tasks such as compound classification. However, without considering the variance of that same feature among controls, conclusions could be misleading. Thus, including both control and treatment samples within the EMD phenotypic profile allows "low activity" features to be identified based on their variance in treatment wells relative to controls. Given that control samples show differing levels of variation among features, we sought to identify "active" features by selecting those with treatment variance at least double the variance of their control counterpart. This procedure resulted in the removal of 23 "inactive" features based on their low relative variance (Supplementary Fig. 7c and Supplementary Data 5, "low variance").

In summary, the extracted cell features were reduced based on four criteria: feature reproducibility among replicates, information content, biological relevance and activity as judged by relative responsiveness to controls. These filtering steps resulted in the reduction of 174 measured features from 11 fluorescent markers by 60% to a final set of 69 features spread across the four assay panels (Supplementary Data 5, "active features").

**Comparison of cytological profiles**. To assess the effectiveness of our data processing and quality control approach (described in "Positional effects adjustment and data standardization" and "Feature reduction") and how the profile might differ under alternative data preparation conditions, we compared the downstream analysis of the profile to two alternative cases. First, we considered the raw unprocessed full-feature data profiles to demonstrate the shortcomings of not correcting for position and plate effects, nor reducing the feature space (Supplementary Fig. 8a, b). For the case of raw unprocessed data, replicate feature distributions are still merged and EMD scores (chemical perturbations) are measured relative to the global control (described in Fig. 5a–c). Second, we compared full-feature profiles with the final reduced 69-feature profiles (Supplementary Fig. 8c) to assess whether global phenotypic differences (i.e., control, low stress and toxic regions) are as clearly revealed (Supplementary Fig. 8d).

The unprocessed data clustergram reveals several different control groups mixed within the treatment clusters (Supplementary Fig. 8a). This suggests that some changes between these treatments and the controls are masked by the technical noise present within the raw data. For both the raw and processed full-feature profiles, the UMAP strongly separates the Brefeldin-a cluster from all other treatments (Supplementary Fig. 8b, c), causing other distinctions to be obscured. The fully processed, feature-reduced profiles (Supplementary Fig. 8d) more clearly separate the treatment groups from controls (particularly the low-stress cluster), and the transitioning patterns of compounds from low to high concentration are more clearly revealed in the UMAPs.

**Statistics and reproducibility**. All statistical analyses were performed using R software[56] and figures were produced using the package ggplot2[57]. The rationale in the data quality control steps was to make use of standard statistical methods to detect and adjust for plate positional effects. Numerical feature data was modeled as a function of two categorical variables (row and column position) using the two-way ANOVA model to assess uniformity among the control wells on each plate[35]. A full summary of the two-way ANOVA analysis including F-statistic and p-value denoted by subscripts r (row) and c (column) is included in Supplementary Data 1. For plates showing non-uniformity in any measured feature, median polish was applied for full plate well-level adjustment[34]. Then individual cells were adjusted for plate positional effects using the well-level adjustment amount. To account for plate-to-plate variation, individual cells were further standardized to the control cells on each plate, using the BZ score. Cell feature distribution plots showing pre- and post- data adjustment and standardization are included in the main text (Fig. 3d) as well as additional supporting figures displaying features under different chemical perturbation conditions (Supplementary Fig. 1d–h).

After data normalization, we compared the sensitivity of three statistical metrics in detecting dissimilarities between pairwise replicate cell populations (Fig. 4 and Supplementary Fig. 2). Statistical tests were carried out by comparing each control well in plate 1 (rep1, rep2, rep3) to its replicate with the same well id on plate 2 (rep1, rep2, rep3), with 9 total comparisons per control well. This resulted in 9 replicate comparisons of 174 feature distributions for each of the 55 control wells using three different statistical metrics. Similarly, each treatment well was compared to its replicate with the same well id on replicate plates (rep1, rep2, rep3), with 3 total comparisons per treatment well, resulting in 3 replicate comparisons of 174 feature distributions for each of the 65 compounds at 7 different concentrations (455 treatment wells). Output summary of replicate reproducibility analysis, as well as metadata (including plate number, well id, and sample size for each statistical test), are included in Supplementary Data 2.

The EMD score was used for profiling phenotypic changes of all treatments (including all DMSO samples) relative to the global control population; raw full-feature EMD profiles are included in (Supplementary Data 6, "raw profile"). The EMD calculation in Fig. 5a (EMD = 2.06, right panel inset) is included in the associated R script as described in Supplementary Table 1. Feature reduction was performed using raw EMD profiles, treatment groups causing strong cell loss within each panel (A, B, C1, and C2), with more than 70% cell reduction relative to the control were identified. Profiles of those treatments (Supplementary Data 6, "toxic treatments") were excluded from the feature reduction analysis steps. The reduced feature EMD profile was log-transformed and features were min-max scaled to the range [0, 1] (Supplementary Data 6, "scaled EMD profile").

Similarity analysis by hierarchical clustering used Euclidean measure to obtain the distance matrix and average linkage method for clustering. The four broad clusters defined in Fig. 6a were identified based on the outermost branches of the dendrogram connecting "similar" treatments (rows). Dimension reduction by UMAP in Fig. 6b is generated using the R package "umap"; cell count (as percent of control) is projected onto the UMAP for visualization and interpretation of phenotypic stress across different dimensions. Phenotypic signatures for all 65 compounds at seven concentrations are provided in the form of radial plots in Supplementary Fig. 9.

## Data availability

All source data underlying the plots and visualizations in this manuscript are available in the GitHub repository at https://github.com/GunsalusPiano/EMD. Data files specific to each figure are summarized in Supplementary Table 1. Any additional supporting data are available upon request.

## Code availability

All R scripts used to generate the plots and visualizations in this manuscript are available in the GitHub repository at https://github.com/GunsalusPiano/EMD. A summary of R scripts used for each figure in the manuscript is available in Supplementary Table 1. Any additional supporting code is available upon request.

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

## Acknowledgements

The authors thank Nikolaos Giakoumidis (NYUAD Core Technology Platforms) for the maintenance of the High-Throughput Screening Platform; Fathima Shaffra Mohammed Refai and Julie Connelly (NYUAD Center for Genomics and Systems Biology) for assistance with experiments; and Paul Selzer (Novartis, Basel, CH), Roger Linington (Simon Fraser University, Vancouver, CA), and Marc Bickle (Roche, Basel, CH) for technical advice and useful discussions. This work was supported by Tamkeen by an NYUAD Research Institute grant to the NYUAD Center for Genomics and Systems Biology (ADHPG-CGSB).

## Author contributions

Study conceptualization: K.C.G., Y.E.P., and S.K.; experiments: S.K.; software and analysis: Y.E.P. and G.L.B.; result interpretation: Y.E.P., X.X., S.K., and K.C.G.; manuscript: Y.E.P., X.X., S.K., and K.C.G.; assay automation: S.K. and H.F.; funding: K.C.G.

## Competing interests

The authors declare no competing interests.
