## [Peer Review File · Communications Biology]

Reviewers' comments:

Reviewer #1 (Remarks to the Author):

The statistical framework for high-content phenotypic profiling using cellular feature distributions

In this manuscript, the authors describe their framework for high content phenotypic profiling. They describe each step and its rationale in detail and demonstrate the performance of their framework by successfully classifying a diverse set of compounds according to their mechanism of action. The field of phenotypic profiling lacks any consensus framework and the authors put forward their framework that could be adopted as the standard framework in the field.

Though the results are encouraging, there are few concerns which need to be addressed before the manuscript can be recommended for publication

Concerns:

Can the authors comment on how the corrections made for removing the technical noise in the data affect the distribution of the features? Since the main focus of their approach is to find differences between the feature distributions, it would be good to know what is the effect of the corrections on the features and if they retain biologically relevant information.

During feature selection, the authors remove one channel completely as the information it contains is captured by another channel. Can the authors comment on the purpose of gathering information from this additional channel? Also, the authors state that their method can be expanded further by measuring additional fluorescent dyes. Can the authors comment on whether it is possible to expand their assay without resulting in redundant channels?

The authors started with 164 features and at the end of feature selection they ended up with 66 features which seemed to be sufficient for distinguishing the diverse MOA in this dataset. Can the authors comment on whether, in general, 164 features will be sufficient for distinguishing compounds and MOA in all datasets?

The authors show several examples of how a compound may affect the cell count or the cell cycle. It would be good to include other examples of phenotypes that can be explained using this approach.

The manuscript will benefit from the authors showing the effect of not correcting for the plate and position effects using downstream analysis and comparing that with what they have in the manuscript.

The authors correct for both plate and well position effects. Can the authors comment on how they would correct for batch effects, which is pertinent especially if their framework is used as a standard in the field and others would like to compare their data against.

Reviewer #2 (Remarks to the Author):

The authors describe a collection of methods for analysis multi-spectral phenotypic profiling experiments, starting from design of staining panels for visualising cellular compartments, including feature extraction and selection strategies, statistical correction of spatial effects, and introducing an interesting distance metric for detecting differences between different feature 'fingerprints' in high-dimensional space.

The paper is generally well written with sufficiently detailed explanations of the methods developed and used, and is attempting to handle some of the long-standing problems and difficulties in the

phenotypic profiling field, namely the ability to discriminate between technical variation and genuine differences in high dimensional feature data, and the ability to utilise more of the information present in single cell distributions.

As the authors describe, the two-way ANOVA and B-score normalisation method for plate correction is not in itself novel, however the application to high-dimensional feature data and correction at the single cell level to enable aggregation of cellular distributions from multiple replicates across wells and plates is an interesting concept. The introduction of the Earth Mover Distance (EMD) metric appears to be a powerful and sensitive method for detecting the magnitude, if perhaps not the direction, of changes to a phenotypic profile due to a perturbation.

Taken together these are interesting contributions to the field, which could perhaps be strengthened by a more thorough treatment of the current literature and the advantages brought. For example, the Cell Painting method mentioned by authors has a reasonable amount of literature around feature extraction, feature selection and robustness and reproducibility metrics, the Pelkmans group has also published a pipeline to perform image-based single cell profiling, and Recursion have also reported a range of methods for high-dimensional data analysis. In the context of these works, could the authors explain the advantages of their proposed approach, for example in terms of improved reproducibility, sensitivity, etc?

In addition, some points of discussion or clarification could add value to the paper:

- One of the big challenges in profiling is batch correction, which would enable comparisons across datasets, cell lines, instruments, etc, and provide more confidence in the reproducibility of a dataset. This approach appears as though it could be useful for correcting this batch-to-batch variability - is this something that the authors have tested and can show reproducibility of features across experiments?
- A potential source of variation in a phenotypic screening pipeline is the choice of parameters in classical cell segmentation and feature extraction, with different software packages and indeed different users producing different feature sets. Can the authors comment on the robustness of the final clusters and trajectories to the choices made at the image analysis stage?
- Along the same lines, by using basic cytoplasm ring segmentation potentially a lot of morphological and cell shape information is being lost. What is the justification for this?
- All phenotypic profiling methods convert image data into feature vectors. The EMD appears to be more sensitive at picking up differences in feature vectors, however, is there a clear improvement in the representation obtained using the EMD fingerprint vs eg straightforward median averaging of features - in other words, are meaningful differences in compound trajectories or clustering found, which would not be found by existing methods?
- The UMAP (Figure 6B) shows the main clusters, but also several smaller clusters and isolated points. It would be useful to have some mechanistic annotation of these clusters, for instance, do the clusters contain compounds with similar annotations, or compounds with close Tanimoto similarities, or simply similar cellular phenotypes? From a statistical point of view, is it possible to understand when two compounds have a similar or a different mechanism of action, ie what is a meaningful value of the EMD metric between two compounds?
- In a similar vein, the benefit of using classically measured cellular features, as opposed to end-to-end deep learning approaches, is a degree of interpretability of the different clusters. Beyond the hand-picked high-level features of cell count and cell cycle shown in figure 7, what are the features which contribute to, for example, the different trajectories in figure 7A, and do these make intuitive sense from a mechanistic perspective (two trajectories for nocodazole and irinotecan are similar at low concentrations and then diverge at high concentrations - is this similarity and divergence apparent from the original images)?
- One might expect that some of the textural and morphological features show greater discriminatory power between treatments, and therefore in figure 7 it might be more impactful to have a visualisation which makes use of the whole fingerprint, rather than hand-defined and standard endpoints (cell count and cell cycle) which do not actually require the profiling approach? What is the motivation for using cell count and cell cycle instead of looking at the fingerprint in an unbiased manner?

- Minor comments/questions:

- line 169 - B score makes assumption that small percentage are active, c.f. Loess. Is this likely to be the case for the selected panel of compounds, and for profiling experiments generally?
- line 492 "dose-dependent phenotypic trajectories assist in discriminating the activity of different compounds" - apparent in umap, but what is a genuine difference vs technical variation? EMD useful for this? Can the many diverse MoAs described be identified using this method?
- EMD is defined to be positive, but the median value fluctuates around 0 in figure 5D, has some correction already been applied here?
- Perhaps I missed this, how cell count integrated with the EMD metric when looking at the dimensionality reduction and clustering?
- clustering - how is the cut-off for finding the 4 clusters determined?
- In the interests of establishing community standards and for wider adoption, it would be useful if the code could be made available via GitHub or similar.

Adam Corrigan
Discovery Sciences, AstraZeneca

Reviewers' comments:

Reviewer #1 (Remarks to the Author):

The statistical framework for high-content phenotypic profiling using cellular feature distributions

In this manuscript, the authors describe their framework for high content phenotypic profiling. They describe each step and its rationale in detail and demonstrate the performance of their framework by successfully classifying a diverse set of compounds according to their mechanism of action. The field of phenotypic profiling lacks any consensus framework and the authors put forward their framework that could be adopted as the standard framework in the field.

Though the results are encouraging, there are few concerns which need to be addressed before the manuscript can be recommended for publication.

Reviewer 1 Comment 1:

Can the authors comment on how the corrections made for removing the technical noise in the data affect the distribution of the features? Since the main focus of their approach is to find differences between the feature distributions, it would be good to know what is the effect of the corrections on the features and if they retain biologically relevant information.

This is an important point, as we didn't want to lose any biologically relevant information during the data adjustment and standardization steps. To summarize our approach (described in the Manuscript lines # 181 - 189, Positional effects adjustment and data standardization), we first inspect each plate (including all features) for the presence of row or column effects using only the median values of the control wells, which are interspersed across all wells and columns. The two-way ANOVA model allowed the detection of any positional effects and was able to distinguish plates/features with and without technical noise (Fig. 3 and Supplementary Figure 1 A-B). It is important to note, however, that a poor plate layout (without sufficient representation of controls in all rows and columns) could hinder the proper identification of technical noise and lead to misinterpretation of technical noise as true perturbations of biological signals.

Plates showing significant technical noise were adjusted by applying median polish, which iteratively computes row and column effects by repeatedly subtracting row and column medians from each well median until the differences become negligible. This residual matrix is then subtracted from the original raw data matrix, which we call the adjustment amount (see Manuscript Fig. 3c). This adjustment is then applied to each well distribution on the plate (raw data for plates that did not show significant technical noise were not subjected to this intermediate step). All well distributions were then standardized to the per-plate control distribution by subtracting the control median and dividing by the MAD of the control wells. These corrections make the distributions across all wells more comparable by putting them all on the same relative scale, but do not affect the shapes of the distributions and thus should not obscure the relative differences between them.

To illustrate this point, cell feature distribution plots showing pre- and post-data adjustment and standardization were included in the main figures of the submitted manuscript (see Fig. 3d) as well as additional supporting figures showing features from other channels (see Supplementary Figure 1 I - K). In this specific case (Fig. 3a & c, plate1 rep1) we observed strong row effects, where rows showed a high/low intensity pattern. This technical noise was manifested in DMSO control cell feature distributions, which separated the wells into two distinct distributions (Supplementary Figure 1 C left). Adjusting the single cell data for this row effect brings the two separated populations closer together (Supplementary Figure 1 C middle), while standardizing the data to the per-plate control cells further controls for plate-to-plate variability (Supplementary Figure 1 C right) and brings the per-well distributions to a common unitless score relative to the control cells on that plate. While these distributions represent control cells (not treatment cells), we confirm that biological information is retained. Here the cell cycle distribution maintains its classical shape showing all phases (G1/S/G2) of the cell cycle after cells were adjusted for positional effects and plate to plate variation.

To address the reviewer's comment, we further illustrate the effects of the median polish and standardization by including additional data panels in Supplementary Figure 1 (in support of Manuscript Fig. 3). Panels (Supplementary Figure 1) D & E show cell feature distributions of chemically treated cells from two replicate plates: Plate 1 rep 3 did not have plate effects and was not adjusted for any technical variation, whereas plate 1 rep1 showed strong row effects that were corrected. In panels (Supplementary Figure 1) F - H, we highlight treatments with Bendamustine (orange, F), LTX-315 (yellow, G), and Rolipram (red, H) to demonstrate how distributions are affected by data correction. In panels F-H, Row 1 shows raw data distributions from different treatment concentrations / wells (gray curves) and control DMSO curve (dashed black curves), and Rows 2 and 3 show the post-adjusted and standardized feature distributions. Both Bendamustine (F) and LTX-315 (G) elicit strong dose-dependent cell cycle effects. Before adjustment, Rolipram (H) distributions look rather heterogeneous due to the row effect, which could manifest as a false positive treatment effect. Once the data are adjusted and standardized, the data show only a minor effect on cell cycle at the concentrations tested.

In summary, we show that data adjustment and standardization can reduce the presence of technical noise, and that all biological effects are preserved in the post-corrected data. While only those plates showing strong technical noise are adjusted using median polish, per-well feature distributions are standardized on all plates to minimize plate-to-plate variation. Furthermore, we were able to preserve the maximum amount of biological information within each treatment group (compound and concentration) across all features by combining standardized replicate cell data from multiple plates, as combining data from multiple wells increases the sample size of each treatment.

Reviewer 1 Comment 2:

During feature selection, the authors remove one channel completely as the information it contains is captured by another channel. Can the authors comment on the purpose of gathering information from this additional channel?

When we designed the HCS assay presented in this manuscript, our priority was to cover a broad spectrum of cellular markers, hence the different panels were designed to maximize the number of cellular compartments we could measure. Only after analyzing

the data did we observe that the signal of the lysosomal channel overlapped to some extent with the peroxisomal channel. We believe this is because the peroxisomal marker is genetically encoded and exhibits a robust signal, whereas the lysosomal marker is a chemical dye whose signal intensity is relatively weak after cell staining and subsequent washing/fixation steps. Although the emission spectra of the two signals overlap only partially, the stronger peroxisomal signal bleeds into the weaker lysosomal signal as a result of the relatively long exposure time needed for the lysosomal marker.

Due to the important roles of lysosomes in the degradation and recycling of cellular waste, cellular signaling and energy metabolism, we remain interested to learn more about the activity of chemical compounds on lysosomes and to integrate this information with activities on other cellular markers. In future studies a genetically encoded lysosomal marker (e.g. mKO2-LAMP1) or a different chemical-fluorophore combination with a stronger signal could replace the lysosomal marker used in this study.

Reviewer 1 Comment 3:

Also, the authors state that their method can be expanded further by measuring additional fluorescent dyes. Can the authors comment on whether it is possible to expand their assay without resulting in redundant channels?

This is an interesting and important question. In designing our assay system, we considered the tradeoff between information content and resource investment (in terms of both time and cost). Our choice of three marker panels with four channels each was designed to survey a majority of cellular organelles and compartments with an array of relatively inexpensive chemical dyes or genetically encoded markers, seeking to balance reagent costs for larger screening applications with the number of panels required (since each additional panel increases the total time and number of plates per experiment).

The number of distinct channels that can be measured using a single marker panel is dictated by the excitation/emission spectra of the fluorophores used, the signal-to-noise ratio, and the capabilities of the microscope setup (type of illumination, bandpass filtering options, and mode of acquisition). As described above, in this study we found that one pair of markers showed overlapping signals due to an unanticipated imbalance in signal intensity in different channels. However, given the wide variety of fluorophores and different chemistries now available, coupled with advancements in instrumentation, it is now feasible to simultaneously image up to 7 or 8 spectrally distinct fluorophores (e.g. see <https://www.frontiersin.org/articles/10.3389/fmicb.2019.01383/full>; <https://www.zeiss.com/microscopy/us/products/confocal-microscopes/lsm-980.html>).

While the current study focuses on cellular morphology by surveying major cellular compartments, many more specialized markers exist that could also be included in multiplexed assay panels. Examples include markers for more in-depth characterization of the cell cycle (e.g. BrdU/EdU, pH3, FUCCI), apoptosis (Annexin V, Caspase 9), DNA damage (H2AX), or components of important signaling pathways (p53, NFkB etc.). Many of these are based on immunofluorescence, which is more cost- and labor-intensive, but offers more spectral flexibility through the choice of fluorophore-tagged secondary antibodies.

Thus, expanding the assay system to include additional markers will simply be an issue of identifying the right combination of instrument design and spectrally distinct fluorophores coupled to chemical or genetic reporters or to secondary antibodies. The number of panels may be tuned depending on the particular application, the number of reporters desired, the level of multiplexing that can be achieved per panel in a particular setup, and the budget of the lab.

Reviewer 1 Comment 4:

The authors started with 164 features and at the end of feature selection they ended up with 66 features which seemed to be sufficient for distinguishing the diverse MOA in this dataset.

Can the authors comment on whether, in general, 164 features will be sufficient for distinguishing compounds and MOA in all datasets?

This is a very good question. We started with 174 features which were reduced to a final set of 69 features. The manuscript previously stated 66 but this was an error which has been amended (refer to manuscript lines 340 and Fig. 6a).

In general, feature selection/reduction leads to improved classification performance for many machine learning tasks since removing unimportant or uninformative features simplifies models and reduces both noise and the risk of overfitting. The exact number of features needed to distinguish between different classes will depend on the nature of the classification problem, the underlying data, and how the data were processed, so there is no easily generalizable rule of thumb for the optimal number of features required for classification. For example, in HCS applications the feature extraction software used (often dictated by proprietary microscope software) will produce different final feature sets, and experimental variables such as chemical library, treatment concentration, and signal detection methods (e.g. reporter assays) will also affect dataset characteristics.

The intent of this study was to focus on data processing and analysis from the early quality control stage to the downstream interpretation of compound activity. We introduced new automated methods for identifying faulty/noisy features, as well as methods for identifying features which do not contribute to the diversity of information we aim to retain (see manuscript line # 279 Feature reduction section). We then grouped/classified compounds based on dose-dependent phenotypic trajectories, and further used dose responses of cell cycle and cell count to support the grouping of compounds into activity classes. Since we did not rely on MOA class annotations for classifying treatments but specifically chose 65 compounds of diverse structure in order to target different cellular components, this dataset would not be an ideal candidate for a machine learning classification task based on shared MOA or target classes. However, we anticipate that the workflow we have developed will be useful for future studies aimed at classification, which is one of the main goals of this field.

To illustrate the effects of reducing the feature space from 174 to 69 using the methods outlined in the manuscript, we have included (Supplementary Figure 5), which compares UMAP trajectories for the full and reduced feature sets. This exercise shows that while the UMAP constructed with the full feature set strongly separates the brefeldin-a cluster (labeled in gray) from the rest of the data, the control and low-stress treatment groups are

largely indistinguishable. Thus selection of a reduced set of informative features allows us to better discern distinct phenotypic classes.

It is important to note however that feature selection will also depend on the method used for quantifying differences in feature distributions between controls and treatments. Different measures (Median, KS, EMD) could result in different final feature sets and thus different profiles.

Reviewer 1 Comment 5:

The authors show several examples of how a compound may affect the cell count or the cell cycle. It would be good to include other examples of phenotypes that can be explained using this approach.

This is a good point. While we do illustrate examples of phenotypic signatures for different compounds in Manuscript Figures 5 & 7, we have now also added a new supplementary figure (Supplementary Figure 6) that further explores phenotypic signatures for two additional compounds with similar but non-identical MOAs (please see response to Reviewer 2, comment 6 and 7). We have also included a supplementary figure showing phenotypic signatures for each compound at all concentrations tested as radial plots (Supplementary Figure 8). Since the current manuscript focuses primarily on the data analysis workflow, further analysis of discriminating features among treatment groups will be the focus of a planned future manuscript.

Reviewer 1 Comment # 6:

The manuscript will benefit from the authors showing the effect of not correcting for the plate and position effects using downstream analysis and comparing that with what they have in the manuscript.

We thank the reviewer for this suggestion. To address this comment, we ran the data workflow again, but this time without adjusting for positional effects and plate to plate variation. Below we provide a side-by-side comparison of the differences in the results, and we now include the clustergram and UMAP results for the raw data as a new supplementary figure (Supplementary Figure 4) which is discussed in the updated manuscript at line 369. **Panel A of figure R1C6** illustrates the differences between the full workflow and the abbreviated workflow for the raw data. The replicate cell feature distributions were still merged to form larger populations, and scored relative to the global control using the EMD scoring method. In panels **B** and **C/D**, we reproduce on the left the figures from the manuscript (hierarchical clustering and UMAPs for EMD profile data after data adjustments and standardization, using the reduced 69 feature set); on the right, we show the corresponding figures for the raw data (without adjusting for positional effects and plate to plate variation, including all 174 features). These comparisons reveal that the processed data more clearly separate the treatment groups from controls (particularly the low stress cluster), and the transitioning patterns of compounds from low to high concentration are more clearly revealed in the UMAPs. While the UMAPs for the raw data show a wide range on DIM2 due to the strong separation of the brefeldin-a cluster from all other treatments, other distinctions are obscured. This suggests that small changes

between these treatments and the controls are masked by the technical noise present in the raw data. Therefore, we believe the data correction step is critical for distinguishing phenotypic changes, which might otherwise be difficult to distinguish from noise.

The authors correct for both plate and well position effects. Can the authors comment on how they would correct for batch effects, which is pertinent especially if their framework is used as a standard in the field and others would like to compare their data against.

Although the chemical treatments for this particular experiment were carried out at the same time, the plates were processed and scanned at different times, leading to some differences in detection efficiency. In principle, the correction for batch effects is the same for plate effects, since each plate contains its own control set and the control set will be subject to the same experimental noise as the treatment samples in the same plate. Thus, adjusting and standardizing each plate to its own control set should account for all the above effects: positional, plate, and batch.

Similar to batch effects, we needed to account for panel to panel variation. To do this we chose to use a common marker (Hoechst33342 for DNA staining and identification of the cell nucleus as the primary object for cell segmentation) within each panel (see figure for Reviewer 2 Comment #2 A). Benchmarking the different panels with a common marker allows us to verify feature reproducibility and integrate fully standardized data from multiple panels and batches (see figure for Reviewer 2 Comment #2 A).

Reviewer #2 (Remarks to the Author):

The authors describe a collection of methods for analysis multi-spectral phenotypic profiling experiments, starting from design of staining panels for visualizing cellular compartments, including feature extraction and selection strategies, statistical correction of spatial effects, and introducing an interesting distance metric for detecting differences between different feature 'fingerprints' in high-dimensional space.

The paper is generally well written with sufficiently detailed explanations of the methods developed and used, and is attempting to handle some of the long-standing problems and difficulties in the phenotypic profiling field, namely the ability to discriminate between technical variation and genuine differences in high dimensional feature data, and the ability to utilize more of the information present in single cell distributions.

As the authors describe, the two-way ANOVA and B-score normalization method for plate correction is not in itself novel, however the application to high-dimensional feature data and correction at the single cell level to enable aggregation of cellular distributions from multiple replicates across wells and plates are interesting concepts. The introduction of the Earth Mover Distance (EMD) metric appears to be a powerful and sensitive method for detecting the magnitude, if perhaps not the direction, of changes to a phenotypic profile due to a perturbation.

Taken together these are interesting contributions to the field, which could perhaps be strengthened by a more thorough treatment of the current literature and the advantages brought. For example, the Cell Painting method mentioned by authors has a reasonable amount of literature around feature extraction, feature selection and robustness and reproducibility metrics, the Pelkmans group has also published a pipeline to perform

image-based single cell profiling, and Recursion have also reported a range of methods for high-dimensional data analysis.

Reviewer 2 Comment #1:

In the context of these works (cell painting, pelkman, recursion) could the authors explain the advantages of their proposed approach, for example in terms of improved reproducibility, sensitivity, etc?

Thank you for this question, which touches upon an important issue and highlights a need for publicly available reference datasets that could be used by multiple groups to perform systematic comparisons of analysis methods. Future community-wide efforts in this direction would greatly benefit the field as a whole.

Due to differences in experimental setup such as plate layout, image analysis software, and staining panels, it is currently difficult to directly compare this work to other published (HCS) data. However, we clarify here how our methods compare to other published works including advantages over others and include additional comparisons of features extracted from other software.

We thank the reviewer for bringing the Pelkman literature to our attention, which we now cite (see manuscript line # 173 citation #36; <https://www.nature.com/articles/s41597-021-00944-5>). While their workflow uses in-house image processing software, as well as different cell lines and probes (total protein, Nascent RNA, PCNA, DAPI), it supports our single cell (feature distribution) standardization method and emphasizes the importance of the cell cycle feature distribution due to its indirect effects on subcellular processes (global RNA production).

As mentioned (see manuscript line 52), the Cell Painting protocol uses a single panel of six markers imaged in five channels, which constrains the number and diversity of cellular features that can be measured. Our priority for this study, however, was to maximize the diversity of features which we achieved by integrating multiple panels of morphological data, thus casting a wider net (10 cellular compartments) of phenotypes in comparison to other published assays such as Cell Painting (refer to manuscript introduction lines 45 - 59).

Regarding feature extraction, free and open software CellProfiler (maintained by the Carpenter lab and also used by Recursion) has several advantages, as it is able to measure a large quantity of features, including complex measurements like Zernike shape features, texture features as well as cell boundaries.

The Cellomics software installed in our lab functions in a similar way (but measures less features per channel when compared to CellProfiler). We assessed the consistency between a subset of data generated from both software packages, and we are able to confirm that Cellomics and CellProfiler yield nearly identical feature distributions (see subset of features figure R2C1, (A) Cellomics features and (B) CellProfiler features).

The central goal and challenge of this study was to establish an analytical protocol for harmonizing the cell feature distributions from multiple panels, and a number of the methods we developed for this work were designed to specifically address reproducibility

and sensitivity. First, we designed a plate layout in order to cope with per-plate positional effects and plate-to-plate variation (which should also address batch effects) (see manuscript 183 - 191). This plate layout allowed us to develop new methods for detecting which plates/features do and do not show significant technical biases (see two-way ANOVA description). This is beneficial because we avoid applying normalization algorithms to plates which don't need it (this could over-correct and disturb the biological signal).

We also compared the performance of the EMD distance metric with metrics used in other studies because we could not directly compare to other published profiles, due to differences in assay parameters. In doing this, we were also able to utilize different statistical scores to assess the reproducibility and reliability of features (see feature selection section in manuscript). As the theme of our study was leveraging cell feature distributions, for all stages of the data workflow, we expected that feature reproducibility could be better assessed by measuring the differences among samples treated with the same perturbation (see reproducibility section in manuscript).

As a general strategy, we proposed merging replicate feature distributions to form larger populations rather than relying on ensemble averages of each replicate. In doing so, we were able to preserve both strong (near toxic) and weak phenotypic changes with more confidence and increased statistical power (see manuscript Fig. 5a and section Phenotypic profiling using the EMD score lines 245 - 251).

Finally, we provide comparisons of our phenotypic profiles with raw data profiles (see response to reviewer 1 comment #4 and #6 and Supplementary Figure 4) as well as the full feature profiles (Supplementary Figure 5) to illustrate the improvements of our data correction and feature reduction approach.

Reviewer 2 Comment #2:

In addition, some points of discussion or clarification could add value to the paper:

One of the big challenges in profiling is batch correction, which would enable comparisons across datasets, cell lines, instruments, etc, and provide more confidence in the reproducibility of a dataset. This approach appears as though it could be useful for correcting this batch-to-batch variability - is this something that the authors have tested and can show reproducibility of features across experiments?

Although this experiment was conducted as a single batch in terms of compound application, the plates for each panel were seeded at different times (since our U2OS line with the genetically encoded marker grows more slowly) and scanned at different times with different markers, resulting in batch-like effects. In principle, the correction for batch effects is the same for plate effects since each plate will contain its own control set and the control set will be subject to the same experimental noise (positional and plate effects) as the treatment samples in the same plate. Adjusting and standardizing the data to the control set on the same plate will account for all of the above effects.

As each panel (A, B, C1/C2) is stained and scanned as an independent batch of plates, we wanted to be able to integrate features from different panels to form a comprehensive phenotypic profile. To do this we chose to use a common marker (Hoechst33342 for DNA

staining and identification of the cell nucleus as the primary object for cell segmentation) within each panel (figure for R2C2 A). Benchmarking the different panels with a common marker also allows us to verify feature reproducibility and integrate fully standardized data from multiple panels and batches (figure for R2C2 B).

We do see that the discrepancies among different experiments and panels are much more harmonized after applying our data workflow. There are still noticeable differences in the overall distribution among the three panels, which we speculate may result from inherent differences between the cell lines used. Overall, we believe that having a common stain/maker over all plates that gives a generally reliable signal (with a preferred feature like Hoechst dye with cell cycle feature) can be key for benchmarking the normalization over different batches or within batches containing multiple panels.

Reviewer 2 Comment #3:

A potential source of variation in a phenotypic screening pipeline is the choice of parameters in classical cell segmentation and feature extraction, with different software packages and indeed different users producing different feature sets.

Can the authors comment on the robustness of the final clusters and trajectories to the choices made at the image analysis stage?

We thank the reviewer for this question. Indeed, a variety of different software packages for cell segmentation and feature extraction are available, depending on the HCS platform used. The data for this manuscript were generated by using the *Thermo Scientific Cellomics Cell Cycle* and *Compartmental Analysis BioApplications* packages, which are part of the *Thermo Scientific Cellomics Arrayscan* HCS platform used in our lab.

We agree that the choice of cell segmentation/feature extraction parameters influences the outcome of features and therefore need to be adapted carefully to the experimental conditions and the cellular markers in use. In our image analysis, we have control images and treatment images, and we always set up the parameters for segmentation and feature extraction by simultaneously looking at both the control and treatment to make sure the parameters for feature extraction (the quantification data) will capture the image difference. After the parameters were optimized, the whole image set was analyzed using the same parameters and did not need to be changed between experiments or batches.

As discussed in response to Reviewer #3 Comment #1, we did assess the consistency between a subset of feature data generated from CellProfiler and Cellomics, and found that they yield nearly identical feature distributions.

Reviewer 2 Comment #4:

Along the same lines, by using basic cytoplasm ring segmentation potentially a lot of morphological and cell shape information is being lost. What is the justification for this?

We agree that cell shape can provide important information on cellular responses, and ideally should be included in HCS studies. Due to technical limitations of the current study, however, this is something we will need to explore in a future study. As mentioned in our response to Reviewer #2 Comment #3, we used the *Thermo Scientific Compartmental Analysis BioApplications* package for cell segmentation and feature extraction. This software package typically uses the nuclear marker as an easily identifiable “primary object”, which allows for proper identification of single cells. In a second step the cytoplasm is defined by a ring mask around the nucleus. The software can also use other markers (e.g. a whole cell stain) as the “primary object”. In some cases, the Plasma Membrane marker (WGA Alexa Fluor 555) or the RNA stain (SYTO14) can be used as a whole cell stain. However, with our particular cell line and experimental protocols (i.e. seeding density/confluency) we couldn't reliably distinguish single cells using these markers, and thus we didn't include cell shape-related features into our analysis. In future studies this problem could be solved by either using fewer cells per well, so that cells are more separated from each other, or by including an additional specific whole cell stain (e.g. Thermo Cell Mask stain). In addition, alternative software packages such as CellProfiler with more advanced cell segmentation algorithms might produce better segmentation results using the dyes used in this study.

Reviewer 2 Comment #5:

All phenotypic profiling methods convert image data into feature vectors. The EMD appears to be more sensitive at picking up differences in feature vectors, however, is there a clear improvement in the representation obtained using the EMD fingerprint vs eg straightforward median averaging of features – in other words, are meaningful differences in compound trajectories or clustering found, which would not be found by existing methods?

Comparing the downstream trajectories of the UMAP under different profiling metrics is a great suggestion. In developing our workflow, we reasoned that since the median measures only the central tendency of a distribution, it is likely to miss other more subtle phenotypic differences between the perturbed and DMSO-control distributions. Indeed, our primary motivation for applying new analytical and profiling methods was due to the challenges we faced in identifying structure and meaningful phenotypic trajectories when using the more common aggregate well average approach. We addressed this issue in the manuscript (Fig. 4 and manuscript line 193, *Statistical metric performance comparison using replicates*) by comparing the performance of different statistical metrics. We showed that overall the EMD score was most sensitive to differences when compared to the KS distance and the robust Z score.

To address the reviewer's concern, in the Reviewer response figures (see figure for R2C5) we show results of a workflow (panel A) using the median average of features, a clustered heatmap (for comparison to Fig. 6a), and the supporting UMAP with the same compound trajectories shown in Fig. 7a. The heatmap (panel B) shows very poor separation of control and treatment samples based on extracted features, and the UMAP (panel C) reflects this poor discrimination. Phenotypic trajectories (panel D) also appear very close together, limiting the ability to distinguish between them.

Reviewer 2 Comment #6:

The UMAP (manuscript figure 6B) shows the main clusters, but also several smaller clusters and isolated points. It would be useful to have some mechanistic annotation of these clusters, for instance, do the clusters contain compounds with similar annotations, or compounds with close Tanimoto similarities, or simply similar cellular phenotypes?

Thank you for this comment. We are currently writing a follow-up manuscript that will address this next phase of the analysis. We note, however, that there are several limitations to this data set. First, we chose a diverse group of compounds with different annotated MOAs that show little structural similarity (see Tanimoto similarity heatmap in figure 2C); therefore, this particular dataset is not ideal for addressing these questions. We also observed that compounds with different MOA may cluster together at some concentrations but not others, thus there is no straightforward way to do mechanistic annotation based on phenotypic profiles at single concentrations. However we think the concentration-dependent phenotypic trajectories revealed in the UMAPs hold promise for mechanistic discrimination, and we will work toward a fuller characterization of this in future studies.

From a statistical point of view, is it possible to understand when two compounds have a similar or a different mechanism of action, i.e. what is a meaningful value of the EMD metric between two compounds?

Thank you for this important question. Ideally, one would need at least several different groups of compounds that share the same target or MOA to investigate whether an EMD score threshold (to discern two compounds with similar/different MOA (for any feature)) can be established. This would enable determination of statistical thresholds for discriminating different MOA clusters. Unfortunately, the set of diverse chemicals used here would not be suitable for this task due to the small set size and lack of known common MOAs among the selected compounds.

In a similar vein, the benefit of using classically measured cellular features, as opposed to end-to-end deep learning approaches, is a degree of interpretability of the different clusters. Beyond the hand-picked high-level features of cell count and cell cycle shown in figure 7, what are the features which contribute to, for example, the different trajectories in figure 7A, and do these make intuitive sense from a mechanistic perspective (two trajectories for nocodazole and irinotecan are similar at low concentrations and then diverge at high concentrations - is this similarity and divergence apparent from the original images)?

Thank you for this interesting comment. We have explored the diverging phenotypes for these two compounds in further detail and discussed the observed phenotypic responses in the manuscript (see section Phenotypic characterization of selected compounds lines 383), and include a new supplementary figure in support of manuscript Fig. 7 (Supplementary Figure 6).

Irinotecan and Nocodazole both target the cell cycle, although they have different (underlying) biochemical mechanisms (directly targeting DNA processing via inhibition of topoisomerase I vs. interfering with microtubules). Their phenotypic similarity at lower concentrations could be due to their mild response to treatment, which we observe in their

phenotypic fingerprints (Supplementary Figure 6A). Each fingerprint at their highest concentration, however, induces increased activity (with larger discrepancies between the two compounds) in several feature channels including Actin/Tubulin, PMG, Mitochondria and Nucleus Texture (Supplementary Figure 6B).

The overall trajectories reflect the different degree of dose-dependent effects of these two compounds, which may be due to differing MOAs. Irinotecan mainly arrested the cell cycle in S phase without strong cytotoxicity as the concentration increased, as evidenced by the DNA content distribution and cell count (cell count did not go below 50% of the controls even at the highest concentration) (Supplementary Figure 6 C and D left panel). In contrast, nocodazole showed a gradual reduction in the G1 peak and increased the relative abundance of the G2/M peak, with observable cytotoxicity at the highest concentration (Supplementary Figure 6 C and D right panel).

Cells treated with irinotecan display a gradual right shift of cellular feature distributions (the staining of different features becomes stronger) with increasing concentrations (Supplementary Figure 6G). This is consistent with the fact that the cells would undergo replication of different organelles and structures in G1-S phase, and the observation that more cells are arrested at S phase at higher concentrations (see also Xu, 2002, <https://www.sciencedirect.com/science/article/pii/S092375341947213X>).

In contrast, we only observed a clear signal increase in the majority of features at the highest concentration of Nocodazole (Supplementary Figure 6F) when there was very strong cell loss. Thus, several features including cell count and cell cycle, clearly induce different responses with increasing concentrations where the UMAP trajectories diverge.

Reviewer 2 Comment #7:

One might expect that some of the textural and morphological features show greater discriminatory power between treatments, and therefore in figure 7 it might be more impactful to have a visualization which makes use of the whole fingerprint, rather than hand-defined and standard endpoints (cell count and cell cycle) which do not actually require the profiling approach? What is the motivation for using cell count and cell cycle instead of looking at the fingerprint in an unbiased manner?

We have found that using a combination of full fingerprints (or full profiles) and cell count/cell cycle data allows for the most detailed description of cellular perturbations.

Indeed whole fingerprints (both control and treatment samples) are our first mode for understanding the comprehensive phenotypic responses to perturbation, as outlined in the manuscript lines 267 - 279 (see manuscript section Phenotypic profiling using the EMD score) and shown in Fig. 5 d - f. Full fingerprints are also included in Fig. 7d, and we include a new supplementary figure showing radial plots for all 65 compounds (see Supplementary Figure 8).

Cell count and cell cycle are two readily and biologically interpretable metrics that integrate multiple cellular responses and processes. Both are based on the nuclear stain (Hoechst33324) which is also used to integrate data across panels and to extract nuclear morphology features.

For example, changes in the percentage of cells in each phase of the cell cycle are an important indicator of effects on specific processes within the cell, e.g. apoptosis, DNA damage, cell cycle checkpoints or polymerization status of microtubules. On the other hand, the distribution of cell cycle phases also indirectly affects other processes in the cell. For example, if a certain perturbation leads to a greater number of cells in G2 phase (suggesting a G2 cell cycle block) this could also lead indirectly to phenotypic responses across multiple cellular features, since cells will not only replicate DNA but also produce more organelles such as mitochondria during the progression of cell cycle before dividing into two cells.

While cell count is not directly integrated into the EMD profile, we found that projecting cell count (as percent of control) onto the UMAP (see manuscript Fig. 6b) reveals a close relationship with the 3rd UMAP dimension. Thus, cell cycle and cell count are important parameters because they both respond to general cell stress and offer a straightforward way of interpreting the profile as a whole.

Minor comments/questions:

Comment 1:

line 169 - B score makes assumption that small percentage are active, c.f. Loess. Is this likely to be the case for the selected panel of compounds, and for profiling experiments generally?

This varies from lab to lab and would depend on the objective of the screen. For this study, we did not depend on a small proportion of compounds being active as the B score is measured relative to the control cells only and should preserve any outlier biological signals using our standardization approach.

Comment 2:

line 492 "dose-dependent phenotypic trajectories assist in discriminating the activity of different compounds" - apparent in umap, but what is a genuine difference vs technical variation?

Technical variation was identified and removed at the initial stages of the data quality control (see manuscript lines 134-191 and manuscript Fig. 3 and Supplementary Figure 1; please also see our response to Reviewer 1, Comment #1 above, which further discusses identification and removal of technical variation). To better illustrate the effect of removing technical variation, we have added additional panels to Supplementary Figure 1 (please refer to our response to Reviewer 1 Comment 6) for a full explanation including side-by-side comparisons of processed and raw data).

Is EMD useful for this? Can the many diverse MoAs described be identified using this method?

The EMD method displayed higher sensitivity in comparison to more standard statistical metrics (KS and Z score), as described in the manuscript (see lines 192 - 238) and shown

in manuscript Fig. 4. We used the EMD distance metric to successfully profile compounds at multiple concentrations, enabling the tracking of perturbation/phenotypes across a wide range of concentrations. Since in this study we specifically chose 65 compounds of diverse structure in order to target different cellular components, our goal was not to classify treatments based on MOA similarity. However, having demonstrated that the EMD is useful for discriminating phenotypic profiles, we believe its application should be useful also for studies whose aim is to classify treatments by MOA similarity, for example phenotypic profiling of a target-selective library of compounds. This will be very interesting to explore in future studies.

Comment 3:

EMD is defined to be positive, but the median value fluctuates around 0 in figure 5D, has some correction already been applied here?

Thank you for bringing this to our attention.

Here we are showing the radial plot of EMD after taking the log and min/max scaling the full profile to [0,1]. In figure 5d the median is fluctuating, but not around zero, it falls between 0.16 and 0.46, which is now reflected properly on the radial plot axis.

The residuals in Figure 5e do naturally fluctuate around median zero with values between (-0.29, 0.45), but for better visualization we added 0.5 (for all radial plot figures) in order to expand the radial plot, resulting in (DMSO) values ranging between 0.21 and 0.95. This has been corrected in the figure legend (see Manuscript figure 5 legend).

Comment 4:

Perhaps I missed this, how cell count integrated with the EMD metric when looking at the dimensionality reduction and clustering?

Cell count is not directly integrated with the EMD metric, but we found that cell count was correlated with the phenotypic trend along one of the UMAP dimensions, as described in the manuscript (see manuscript lines 354 - 364 and figure 6B). The UMAP in figure 6B is color coded by cell count (via percent of control) and highlights the relationship between cell count and concentration-dependent phenotypic movement across the UMAP induced by treatment with different compounds. Specifically, UMAP dimension 3 discriminates phenotypically active and toxic treatments from the low activity ("low stress") and control groups, as indicated by the decrease in cell counts from top to bottom.

Comment 5:

clustering - how is the cut-off for finding the 4 clusters determined?

The four broad clusters defined in manuscript Fig. 6 a (hierarchical clustering) were identified based on the outermost branches of the dendrogram connecting "similar" treatments (rows). For example, Cluster 1 (yellow) is the DMSO-control group, which separates from the rest of the treatments (blue) and also shows the least amount of

structure within the group. In contrast, Cluster 4 is the most distant from the control and has the highest phenotypic activity (seen as bright green in the heatmap). The samples within Cluster 4 correspond to the toxic or near toxic region of the UMAP as well, whereas Cluster 2 corresponds to the low activity/low stress group of treatments (also visible in the UMAP).

Comment 6:

In the interests of establishing community standards and for wider adoption, it would be useful if the code could be made available via GitHub or similar.

Yes, we agree! R scripts and data files in support of this work can be found at the following Github link, which is now also referenced in the manuscript.

<https://github.com/GunsalusPiano/EMD>

Adam Corrigan
Discovery Sciences, AstraZeneca

A.

Summary of data processing

Figure R1C6: Raw data profiles

(A) Summary of the abbreviated workflow for computing raw data profiles (see steps 4 and 6). Comparison of similarity by hierarchical clustering analysis (B) and dimension reduction by UMAP (C – D) between fully processed data (see left panels) and the raw data (see right panels).

B.

Figure R1C6: Raw data profiles

(A) Summary of the abbreviated workflow for computing raw data profiles (see steps 4 and 6). Comparison of similarity by hierarchical clustering analysis (B) and dimension reduction by UMAP (C – D) between fully processed data (see left panels) and the raw data (see right panels).

C.

Figure R1C6: Raw data profiles

(A) Summary of the abbreviated workflow for computing raw data profiles (see steps 4 and 6). Comparison of similarity by hierarchical clustering analysis (B) and dimension reduction by UMAP (C – D) between fully processed data (see left panels) and the raw data (see right panels).

D.

Figure R1C6: Raw data profiles

(A) Summary of the abbreviated workflow for computing raw data profiles (see steps 4 and 6). Comparison of similarity by hierarchical clustering analysis (B) and dimension reduction by UMAP (C – D) between fully processed data (see left panels) and the raw data (see right panels).

A. Cellomics

B. Cell Profiler

Figure R2C1: Image analysis software

Comparison of feature distributions derived from (A) Cellomics and (B) Cell Profiler image analysis software

A. Cellomics

RNA stain

Mitochondria stain

B. Cell Profiler

RNA stain

Mitochondria stain

Figure R2C1: Image analysis software

Comparison of feature distributions derived from (A) Cellomics and (B) Cell Profiler image analysis software

A.

B.

Figure R2C2: Data integration

(A) Each panel of fluorescent dyes uses a common Hoechst33342 marker. (B) Illustration of pre and post data data standardization of the cell cycle feature from each panel. Cell are treated with control-DMSO.

Figure R2C5: Phenotypic profile comparisons

(A) Workflow for median averaging of merged replicates (B) Similarity by hierarchical clustering of the phenotypic profile, treatments (yellow), control (blue). (C) UMAP of phenotypic profile based on median averages per treatment condition and (D) Phenotypic trajectories for comparisons with manuscript figure 7.

C. UMAP median profile

D. UMAP median profile trajectories

Figure R2C5: Phenotypic profile comparisons

(A) Workflow for median averaging of merged replicates (B) Similarity by hierarchical clustering of the phenotypic profile, treatments (yellow), control (blue). (C) UMAP of phenotypic profile based on median averages per treatment condition and (D) Phenotypic trajectories for comparisons with manuscript figure 7.

REVIEWERS' COMMENTS:

Reviewer #1 (Remarks to the Author):

The authors have satisfactorily addressed my concerns. I recommend the manuscript for publication.